# In vivo reconstitution finds multivalent RNA–RNA interactions as drivers of mesh-like condensates

Weirui Ma[1][†][*], Gang Zhen[1], Wei Xie[2], Christine Mayr[1][*]

[1]Cancer Biology and Genetics Program, Memorial Sloan Kettering Cancer Center, New York, United States; [2]Structural Biology Program, Memorial Sloan Kettering Cancer Center, New York, United States

**Abstract** Liquid-like condensates have been thought to be sphere-like. Recently, various condensates with filamentous morphology have been observed in cells. One such condensate is the TIS granule network that shares a large surface area with the rough endoplasmic reticulum and is important for membrane protein trafficking. It has been unclear how condensates with mesh-like shapes but dynamic protein components are formed. In vitro and in vivo reconstitution experiments revealed that the minimal components are a multivalent RNA-binding protein that concentrates RNAs that are able to form extensive intermolecular mRNA–mRNA interactions. mRNAs with large unstructured regions have a high propensity to form a pervasive intermolecular interaction network that acts as condensate skeleton. The underlying RNA matrix prevents full fusion of spherical liquid-like condensates, thus driving the formation of irregularly shaped membraneless organelles. The resulting large surface area may promote interactions at the condensate surface and at the interface with other organelles.

**\*For correspondence:**
maweirui@zju.edu.cn (WM);
mayrc@mskcc.org (CM)

**Present address:** [†]Life Sciences Institute, Zhejiang University, Hangzhou, China

**Competing interests:** The authors declare that no competing interests exist.

## Introduction

Despite lacking a surrounding lipid membrane, membraneless organelles are micron-sized structures that compartmentalize the subcellular space to organize biological reactions (*Hyman et al., 2014*; *Banani et al., 2017*). Most known membraneless organelles are spherical and include stress granules, P granules, and the nucleolus (*Seydoux and Fire, 1994*; *Kedersha et al., 1999*; *Brangwynne et al., 2009*; *Brangwynne et al., 2011*). Recently, several mesh-like condensates were found in cells that include the TIS granule network, FXR1 condensates, and localization bodies (L-bodies) (*Ma and Mayr, 2018*; *Neil et al., 2020*; *Smith et al., 2020*).

TIS granules are formed through assembly of the RNA-binding protein TIS11B. TIS granules have tubule-like structures and generate a reticular meshwork that is intertwined with the rough endoplasmic reticulum (ER), one of the major sites of protein translation. The mesh-like morphology of the TIS granule network allows it to share a lot of surface area with the ER, thus, generating a large condensate–organelle interface. TIS granules are present in various cell types under physiological conditions and provide a translation environment for mRNAs that contain several AU-rich elements in their 3′ untranslated regions (3′UTRs). Translation in the TIS granule–ER interface, the so-called TIGER domain, allows membrane proteins to form specific protein complexes, indicating that TIS granules are important for the trafficking of plasma membrane proteins (*Ma and Mayr, 2018*; *Mayr, 2019*).

In addition to mammalian cells, mesh-like condensates also exist in other organisms. During *Xenopus* oocyte maturation, the ER and maternal mRNAs localize to the vegetal pole, which is critical for proper embryonic patterning (*Deshler et al., 1997*; *Neil et al., 2020*). One of the major mRNAs that localizes to the vegetal pole in a 3′UTR-dependent manner is *Vg1*. Its 3′UTR is AU-rich and is bound

by the RNA-binding protein Vera (*Deshler et al., 1997*). Condensates that contain *Vg1* mRNA and several RNA-binding proteins, including Vera, were recently discovered in frog oocytes. These so-called L-bodies localize to the vegetal pole and have a mesh-like shape (*Neil et al., 2020*). Although TIS granules were found in somatic cells, they seem to share a lot of characteristics with L-bodies.

Many biomolecular condensates or RNA granules contain protein and RNA (*Teixeira et al., 2005*; *Schwartz et al., 2013*; *Smith et al., 2016*; *Banerjee et al., 2017*; *Fuller et al., 2020*; *Lee et al., 2020*; *Fernandes and Buchan, 2020*; *Hyman et al., 2014*; *Banani et al., 2017*). It is well established that both components contribute to the multivalency of phase separation systems. Through use of RNA, including total RNA (*Van Treeck and Parker, 2018*), RNA with repeats (*Jain and Vale, 2017*), artificial RNAs (*Lin et al., 2015*; *Ries et al., 2019*) or homopolymers (*Elbaum-Garfinkle et al., 2015*; *Wei et al., 2017*; *Boeynaems et al., 2019*), and a handful of mRNAs, it has been demonstrated that RNA can phase separate without protein and that RNA can promote or inhibit phase separation (*Jain and Vale, 2017*; *Maharana et al., 2018*). It has also been shown that protein–RNA interactions can influence the identity and material properties of condensates in vitro and in vivo (*Elbaum-Garfinkle et al., 2015*; *Zhang et al., 2015*; *Langdon et al., 2018*; *Boeynaems et al., 2019*). However, as RNA is not a uniform entity, it is currently unknown how different RNAs influence various aspects of phase separation (*Van Treeck and Parker, 2018*; *Jain and Vale, 2017*; *Lin et al., 2015*; *Ries et al., 2019*; *Elbaum-Garfinkle et al., 2015*; *Wei et al., 2017*; *Boeynaems et al., 2019*; *Zhang et al., 2015*; *Langdon et al., 2018*).

Early on, it was established that spherical condensates are liquid-like (*Brangwynne et al., 2009*). Moreover, different phase separation systems can have a wide range of material properties ranging from liquid-like to gel-like to solid (*Alberti et al., 2019*). Surface tension promotes sphere formation of liquid droplets (*Hyman et al., 2014*). This led to the widely accepted conclusion that non-spherical and irregularly shaped condensates are aggregates and must be gel-like or solid (*Molliex et al., 2015*; *Lin et al., 2015*; *Qamar et al., 2018*; *Boeynaems et al., 2019*). Our recent finding contradicts this notion. In the course of our studies, we made the intriguing and, at first glance, paradoxical observation of condensates with mesh-like or filamentous morphology, but with dynamic protein components. This raised the question of how liquid-like, but non-spherical, organelles are generated and how their mesh-like morphology is determined.

We used chimeric RNA-binding proteins as models to study how the three-dimensional organization of mesh-like condensates is controlled and found that it is determined by RNA. We examined the influence of 47 human in vitro transcribed 3'UTRs on phase separation behavior of an RNA-binding protein. Whereas the addition of predominantly structured mRNAs generated spherical condensates, the addition of largely unstructured mRNAs induced formation of mesh-like condensates. In vitro and in vivo reconstitution experiments revealed that large unstructured RNA regions form extensive, multivalent intermolecular RNA–RNA interactions that drive the generation of mesh-like condensates whose protein components are highly mobile. The non-spherical geometry allows the dynamic membraneless organelles to maximize their surface area. This morphology may promote reactions that occur on the surface or in the interface between RNA granules and other organelles as is the case for TIS granules and the ER.

## Results

### RNA is required to generate mesh-like RNA granules in cells

TIS granules have a mesh-like morphology and form through assembly of the RNA-binding protein TIS11B (*Ma and Mayr, 2018*). When we initially discovered TIS granules, we expressed mCherry-tagged TIS11B in HeLa cells and observed that the transfected TIS granules largely recapitulated the mesh-like three-dimensional structure of endogenous TIS granules (*Figure 1A*). At the time, we performed fluorescence recovery after photobleaching (FRAP) after 16 hr of transfection and observed 20% of fluorescence recovery in 120 s (*Ma and Mayr, 2018*). It is an accepted fact that surface tension drives liquid-like phase-separated condensates to adopt sphere-like shapes as spheres have a minimal surface area for a given volume (*Hyman et al., 2014*). Therefore, we initially thought that TIS granules are mesh-like because they are gel-like (*Ma and Mayr, 2018*).

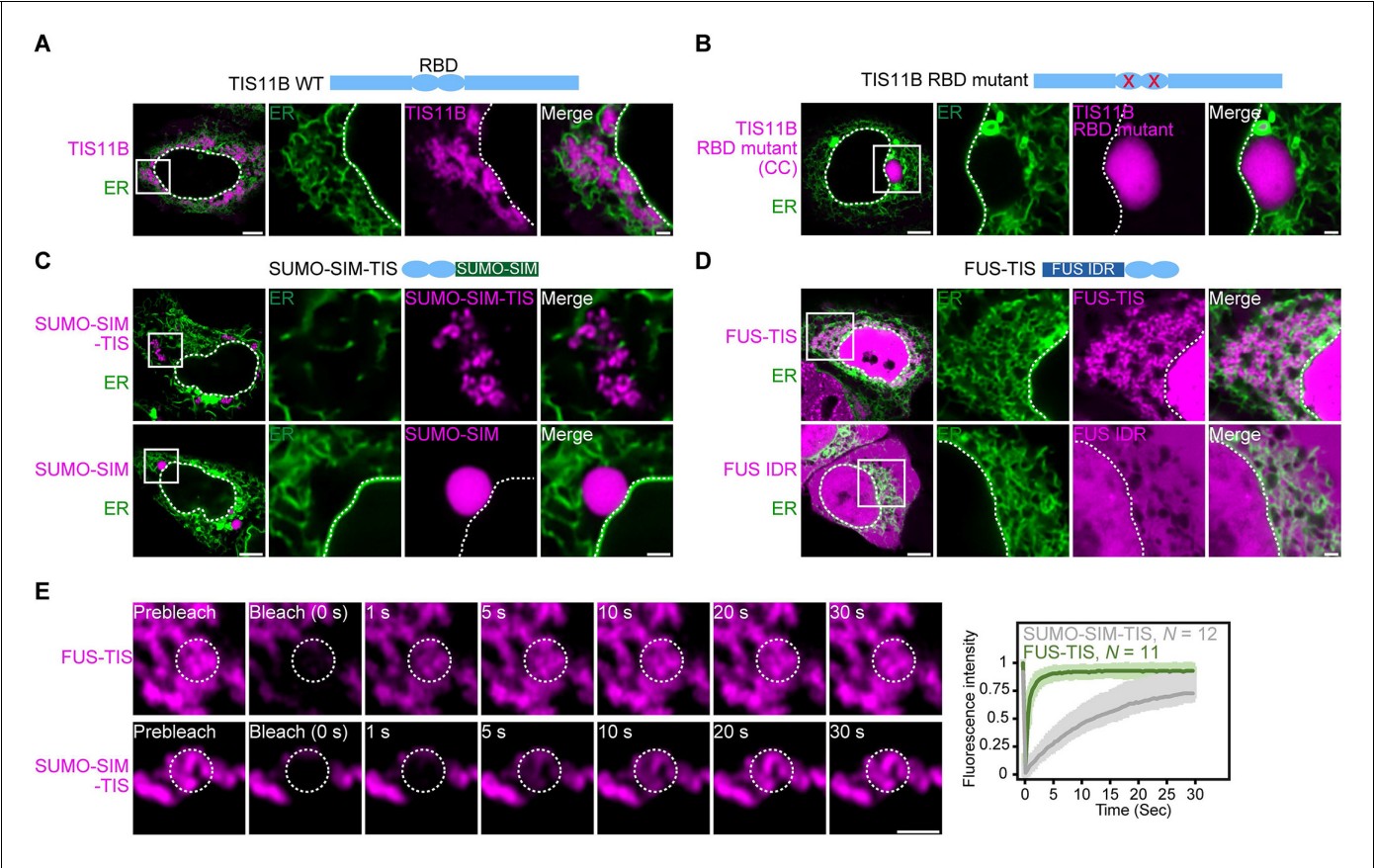

**Figure 1.** RNA determines the morphology of dynamic mesh-like RNA granules in cells. (**A**) Confocal live-cell imaging of HeLa cells after the transfection of mCherry-tagged TIS11B. GFP-SEC61B was co-transfected to visualize the endoplasmic reticulum. The white dotted line demarcates the nucleus. Right: higher magnification of the indicated region. Scale bars, 5 µm (overview) and 1 µm (zoom-in). (**B**) Same as (**A**), but after transfection of TIS11B with a mutated RNA-binding domain. See *Figure 1—figure supplement 1* for more mutants. CC: C135H/C173H. (**C**) Same as (**A**), but after transfection of mCherry-tagged SUMO10-SIM5 or SUMO-SIM-TIS chimera. 73% (*N* = 52) of SUMO-SIM-TIS granules are mesh-like. (**D**) Same as (**A**), but after transfection of mGFP-tagged FUS IDR (amino acids 1–214) or FUS-TIS chimera. All granules are mesh-like. (**E**) Fluorescence recovery after photobleaching of FUS-TIS and SUMO-SIM-TIS 16 hr after transfection of mCherry-SUMO-SIM-TIS or mGFP-FUS-TIS into HeLa cells. Scale bar, 1 µm. The online version of this article includes the following figure supplement(s) for figure 1:

**Figure supplement 1.** Mutation of the TIS11B RNA-binding domain generates sphere-like granules in cells.

**Figure supplement 2.** In the context of various multivalent domains, the TIS11B RNA-binding domain generates mesh-like condensates in vivo.

More recently, we repeated the FRAP experiment at 5 hr after transfection and observed 43% fluorescent recovery of GFP-tagged TIS11B in 10 s (*Figure 1—figure supplement 1A*). This observation suggested that at this time point TIS granules generated from transfected constructs have somewhat dynamic protein components despite being irregularly shaped. This observation is reminiscent of L-bodies in frog oocytes, which have a mesh-like morphology but dynamic protein components (*Neil et al., 2020*). Our FRAP experiments suggested that the mesh-like shape of TIS granules obtained by transfection cannot simply be explained by gel-like biophysical properties. It is important to point out that the material properties of endogenous TIS granules are currently unknown as all experiments so far have been performed with fluorescently tagged TIS11B constructs (*Ma and Mayr, 2018*).

As TIS11B binds to AU-rich elements and the localization element of the *Vg1* mRNA is highly AU-rich, we hypothesized that mRNAs play a role in determining the mesh-like morphology of condensates. TIS11B contains a double zinc finger RNA-binding domain (RBD). When we introduced different point mutations to disrupt RNA binding (*Figure 1—figure supplement 1B–C*; *Lai et al., 2000*), the mesh-like assemblies were turned into sphere-like condensates that are no longer intertwined

with the ER (*Figure 1B*, *Figure 1—figure supplement 1D, E*). This suggested that the mesh-like organization of TIS granules requires the recruitment of mRNAs.

## Mesh-like condensates in cells have highly mobile protein components

Expression of the TIS11B RBD alone is not sufficient for condensate formation (*Figure 1—figure supplement 2A*). To assess the importance of the TIS11B RBD for mesh-like condensate formation, we tested if it results in network formation in the context of different multivalent domains. Expression of SUMO-SIM generates sphere-like condensates in the cytoplasm (*Figure 1C*; *Banani et al., 2016*). However, when we fused SUMO-SIM to the TIS11B RBD (SUMO-SIM-TIS), we observed a filamentous condensate that is not intertwined with the ER (*Figure 1C*, *Figure 1—figure supplement 2B*). This was an important result as it demonstrates that intertwinement with the ER is not necessary for mesh-like condensate formation.

Another well-studied multivalent domain is the intrinsically disordered region (IDR) of FUS (*Kato et al., 2012*). Expression of the FUS-IDR alone did not generate condensates (*Figure 1D*). When we fused the RBD of TIS11B to the IDR of FUS to generate FUS TIS, we observed cytoplasmic mesh-like condensates that look very similar to wild-type TIS granules (*Figure 1D*). In the context of FUS-TIS, the RBD of TIS11B is functional as it recruits the same mRNAs to the condensates (*Figure 1—figure supplement 2C*).

Importantly, both FUS-TIS and SUMO-SIM-TIS show fast fluorescence recovery in FRAP experiments (*Figure 1E*), indicating that the filamentous networks have highly mobile protein components and are not aggregates. These experiments reveal that in the context of various multivalent domains the TIS11B RBD generates mesh-like condensates in vivo. Based on the current knowledge, this observation represents a paradox as it is thought that biomolecular condensates with dynamic protein components should be sphere-like because of surface tension (*Hyman et al., 2014*). However, the seemingly liquid-like condensates generated by FUS-TIS and SUMO-SIM-TIS have mesh-like morphologies in cells. This observation motivated us to investigate how filamentous but liquid-like condensates are generated. To do so, we chose to perform in vitro reconstitution experiments with FUS-TIS as a model system as it formed mesh-like condensates and showed a highly dynamic behavior in the FRAP experiments in cells.

## Specific RNAs drive mesh-like condensate formation in vitro

We recombinantly expressed and purified FUS-TIS (*Figure 2—figure supplement 1A–C*). FUS-TIS phase separates into sphere-like condensates that are liquid-like (*Figure 2—figure supplement 1D, E*). We then added to FUS-TIS in vitro transcribed 3′UTRs of mRNAs, which were recruited to FUS-TIS condensates (*Figure 2A*). The addition of the *FUS* 3′UTR did not change the morphology of the sphere-like condensates formed by FUS-TIS (*Figure 2A*). However, when we added 3′UTRs of TIS11B target mRNAs, including *CD47*, *CD274* (PD-L1) or *ELAVL1* (HuR) (*Ma and Mayr, 2018*), we observed formation of mesh-like FUS-TIS condensates (*Figure 2A*). Importantly, the mesh-like condensates do not represent aggregates as FUS-TIS protein showed fast fluorescence recovery in FRAP experiments performed at 2 and 16 hr after induction of phase separation (*Figure 2B*, *Figure 2—figure supplement 1F*).

All phase separation experiments were performed at two time points (after 2 and 16 hr of incubation) in the presence of 5% dextran and at RNA concentrations spanning three orders of magnitude. Network formation was already observed at the early time point, but longer incubation led to formation of a more extensive network (*Figure 2A, C*, *Figure 2—figure supplements 1G* and *2A–D*). Although the minimum RNA concentration required to induce network formation varied, these experiments revealed that the capacity for network formation is an intrinsic property of the RNA as sphere-forming RNAs did not form networks even at high concentrations. Instead, at high RNA concentrations, we often observed inhibition of phase separation, as was observed previously (*Figure 2C*, *Figure 2—figure supplement 1G*; *Maharana et al., 2018*).

The three network-forming RNAs are longer than the sphere-forming RNA (*Figure 2A*). To examine if network formation is only accomplished by long RNAs, we tested 19 additional RNAs with a length spanning 500–3000 nt. All longer RNAs formed networks, but we observed both network and sphere formation for RNAs shorter than 2000 nt, indicating that network formation is not only determined by the length of the RNA (*Figures 2D* and *3A*, *Figure 3—figure supplements 1A–C* and

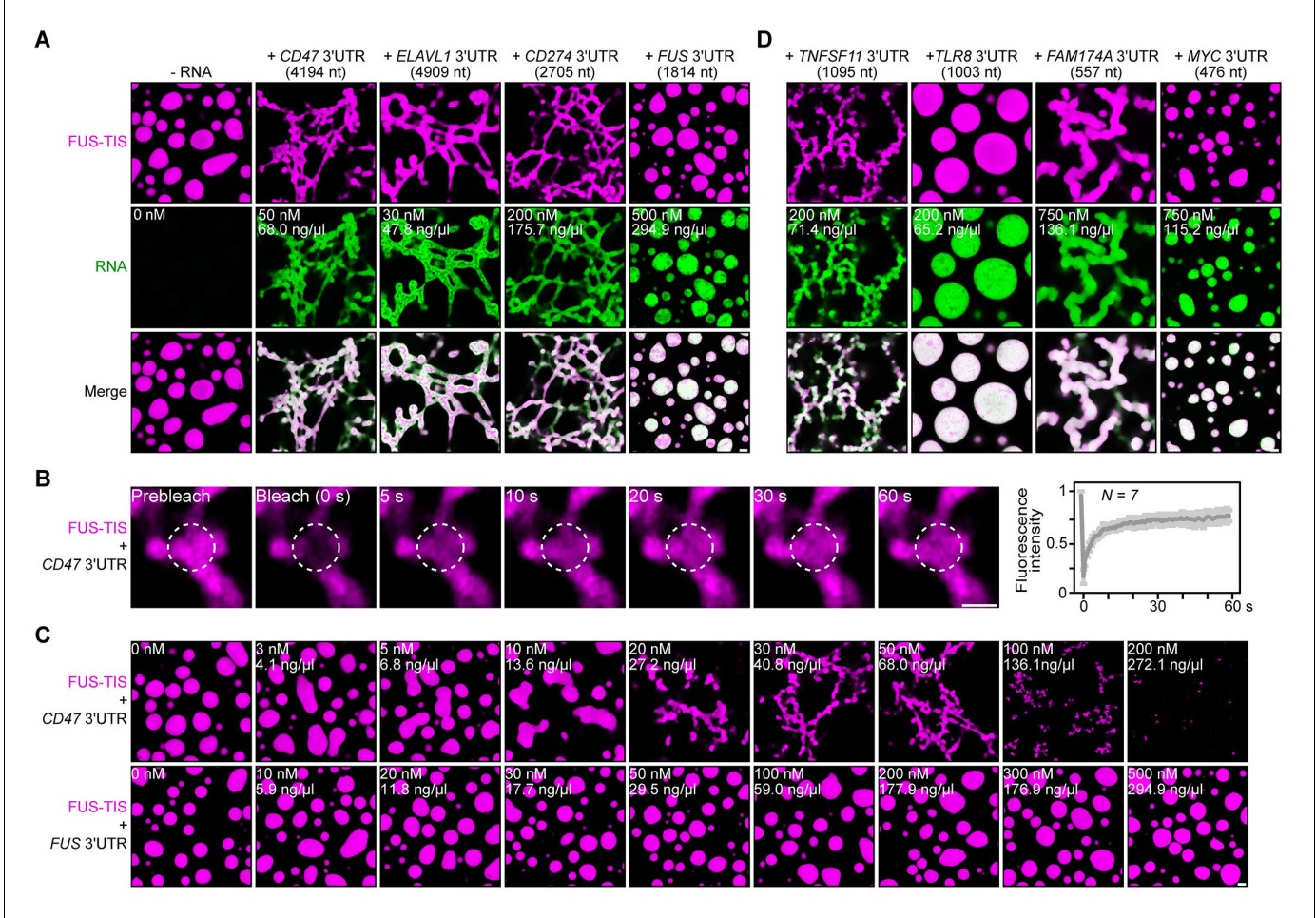

**Figure 2.** Specific RNAs induce formation of dynamic mesh-like condensates in vitro. (**A**) Representative confocal images of phase separation experiments using purified mGFP-FUS-TIS (10 µM) in the absence or presence of the indicated in vitro transcribed RNAs after 16 hr of incubation. Scale bar, 2 µm. Five percent dextran was added into the phase separation buffer as crowding agent in all experiments. (**B**) Fluorescence recovery after photobleaching of mGFP-FUS-TIS (10 µM) mixed with *CD47* 3'UTR (50 nM) after 2 hr of incubation. Scale bar, 1 µm. (**C**) Same as (**A**), but in the presence of different RNA concentrations. (**D**) Same as (**A**), but additional RNAs are shown.

The online version of this article includes the following figure supplement(s) for figure 2:

**Figure supplement 1.** Specific RNAs induce mesh-like condensates in vitro.

**Figure supplement 2.** Specific RNAs induce mesh-like condensates in vitro.

*2A*, *Figure 3—source data 1*). We did not observe phase separation when using high concentrations of the RNAs alone without the addition of FUS-TIS protein (*Figure 3—figure supplement 2B*).

## RNAs that are predicted to have large disordered regions have a high propensity to induce network formation

To identify the responsible determinants for network formation, we focused on the 18 3'UTRs that were shorter than 2000 nt and correlated their ability for network formation with several parameters. Within this size-restricted cohort, the number of AU-rich elements or the GC-content of the RNA had no influence on network formation (*Figure 3B, C*). We then used RNAfold to predict the secondary structure of the RNAs. RNAfold predicts the minimum free energy secondary structure and the centroid structure, which is the RNA structure that contains a minimal base-pair distance to all structures in the thermodynamic ensemble (*Gruber et al., 2008*). Every RNA is predicted to have unstructured regions (indicated by the green and blue colors) and regions of strong local structure (red color code; *Figure 3D, E*, *Figure 3—figure supplement 3A–D*; *Aw et al., 2016*). We observed a strong association between the propensity of an RNA to induce network formation and the

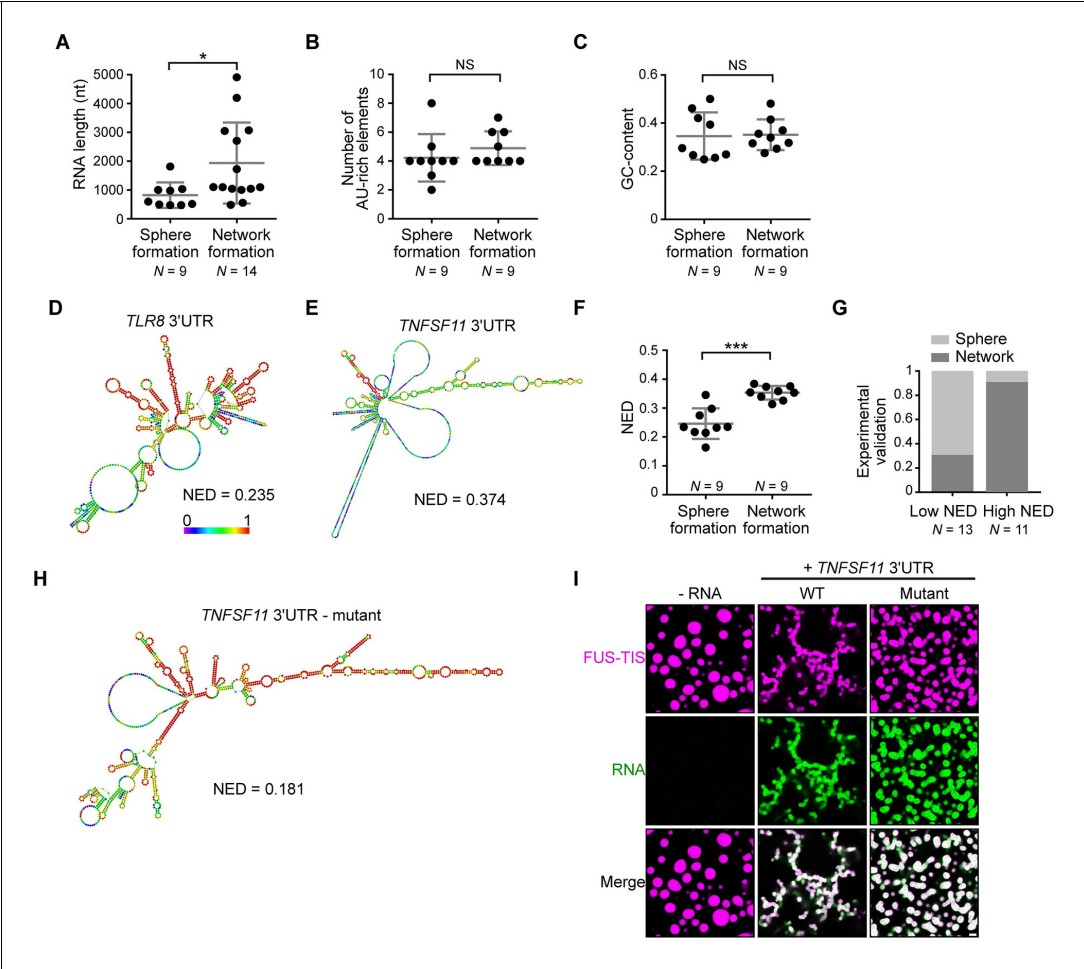

**Figure 3.** RNAs predicted to have large disordered regions have a high propensity to induce network formation in vitro. (**A**) Distribution of length of sphere- and network-forming RNAs. Mann–Whitney test, $Z = -2.76$, p=0.004. See also *Figure 3—source data 1*. (**B**) Number of AU-rich elements in sphere- and network-forming RNAs with a length shorter than 2000 nt. See also *Figure 3—source data 1*. Mann–Whitney test, $Z = 0.190$, p=0.258, NS, not significant. (**C**) Distribution of GC-content of sphere- and network-forming RNAs with a length shorter than 2000 nt. See also *Figure 3—source data 1*. Mann–Whitney test, $Z = 0.566$, p=0.605. (**D**) Centroid RNA secondary structure of *TLR8* 3'UTR predicted by RNAfold. The color code represents base-pairing probability. (**E**) Same as (**D**), but the *TNFSF11* 3'UTR is shown. (**F**) Normalized ensemble diversity (NED) values of sphere- and network-forming RNAs. See *Figure 3—source data 1*. Mann–Whitney test, $Z = -3.3$, ***p<0.0003. (**G**) Experimental validation of $N = 24$ in vitro transcribed RNAs whose ability for network formation was predicted by NED. Sphere formation is indicated in dark gray, whereas network formation is indicated in light gray. See *Figure 3—source data 1*. Mann–Whitney test was performed on the experimental validation, $Z = -2.8$, ***p=0.004. (**H**) Same as (**D**), but the mutant *TNFSF11* 3'UTR is shown. (**I**) Representative confocal images of phase separation experiments using purified mGFP-FUS-TIS (10 µM) in the presence of 150 nM of the indicated in vitro transcribed RNAs after 16 hr of incubation. Scale bar, 2 µm.

The online version of this article includes the following source data and figure supplement(s) for figure 3:

**Source data 1.** Length, number of AU-rich elements, GC-content, and normalized ensemble diversity values of the 47 experimentally tested 3'UTRs.

**Figure supplement 1.** Specific RNAs with various lengths induce mesh-like condensates in vitro.

**Figure supplement 2.** RNA alone does not induce phase separation in vitro.

**Figure supplement 3.** Predicted RNA secondary structures and their corresponding normalized ensemble diversity values for examples of sphere-forming, network-forming, and highly structured RNAs.

**Figure supplement 4.** The normalized ensemble diversity (NED) value of RNAs is highly predictive for their ability to form sphere- or mesh-like condensates.

**Figure supplement 5.** In a size-restricted dataset, the number of AU-rich elements does not predict mesh-like condensate formation.

---

predicted 'unstructured-ness' of the RNA. For sphere-inducing RNAs, the majority of their nucleotides are predicted to form strong local structures, whereas RNAs with a high propensity for network formation are predicted to contain large, unstructured regions (*Figure 3D, E, Figure 3—figure*

supplement 3A–D, *Figure 3—source data 1*). We call the unstructured regions 'large disordered regions' (LDRs) of mRNAs.

Can we use RNA-fold-based structure prediction as a tool to identify network-forming RNAs? Ensemble diversity is the number of potential RNA structures that are predicted for a given RNA (*Lorenz et al., 2011*). As ensemble diversity increases with RNA length (*Ding et al., 2005*), we are using a length-normalized value (NED). RNAs with low NED values have predominantly strong local structures, whereas RNAs predicted to have high NED values often have LDRs (*Figure 3D, E*, *Figure 3—figure supplement 3A–D*, *Figure 3—source data 1*). The NED values correlated strongly with the ability of an RNA to induce network formation and clearly separated the two groups of RNAs with respect to network formation (*Figure 3F*). The majority of network-forming RNAs had NED values that were higher than 0.280, whereas the majority of sphere-forming RNAs had NED values lower than 0.265. This is consistent with their stronger secondary structure as, for example, highly structured RNAs such as tRNAs and six MS2 repeats have NED values of 0.04 and 0.17, respectively (*Figure 3—figure supplement 3E, F*). We want to emphasize that we do not use RNA-fold to predict the correct secondary structure of an RNA molecule, but we are using the NED value that estimates the conformational heterogeneity of an RNA as a tool to predict RNAs that induce formation of sphere-like or mesh-like condensates.

To test the predictive value of NED, we chose a new set of 24 AU-rich element-containing 3'UTRs purely based on their NED values and tested their network-forming abilities. We found that 19/24 (79%) of the tested RNAs were predicted correctly with respect to their sphere- or network-forming abilities (*Figure 3G, Figure 3—figure supplement 4A, B*, *Figure 3—source data 1*). As the number of AU-rich elements in both groups is comparable (*Figure 3—figure supplement 5A*), the high success rate strongly suggests that LDRs of 3'UTRs determine network formation. To test this prediction experimentally, we performed a loss-of-function experiment. We used the *TNFSF11* 3'UTR that contains several LDRs (*Figure 3E*) and introduced strong local base-pairing by the addition of two oligonucleotides that were perfectly complementary to upstream regions and did not disrupt AU-rich elements (*Figure 3—figure supplement 5B*). The *TNFSF11* 3'UTR mutant has stronger local structures indicated by increased base-pairing and a lower NED value, and it has largely lost the ability for network formation (*Figure 3H, I*). Taken together, these results support a model wherein a high diversity of predicted structural conformations correlates with the extent of LDRs in RNAs and is associated with the formation of mesh-like condensates.

## RNAs work additively to induce formation of granule networks

The minimum RNA concentration for network formation was 20 nM. It was observed for the *CD47* and *ELAVL1* 3'UTRs and corresponds to 27 and 32 ng/µl, respectively (*Figure 2C*, *Figure 2—figure supplement 1G*). This is higher than the mRNA concentration in the cytoplasm of mammalian cells, which was estimated to be 8 pM to 8 nM (9.5 pg/µl to 9.5 ng/µl; see Materials and methods) (*Chen et al., 2015*; *Chen et al., 2016*; *Maharana et al., 2018*). As TIS granules and L-bodies contain many mRNAs (*Ma and Mayr, 2018*; *Neil et al., 2020*), we hypothesized that multiple RNAs together may contribute to network formation. Therefore, we tested whether two RNAs co-localize in the network. Labeling of several pairs of RNAs with two different fluorescent dyes showed that they co-localize (*Figure 4A*). Furthermore, the mixing of suboptimal amounts of four network-forming RNAs, together with FUS-TIS, resulted in network formation, indicating that the different RNAs have an additive effect (*Figure 4B*). Importantly, mixing 10 mRNAs together with FUS-TIS only required 2 nM (0.7–3.2 ng/µl) of each mRNA which is substantially lower than the average mRNA expression in cells (*Figure 4B*). These data indicate that the RNA concentrations used for the in vitro experiments are in a range that is physiologically relevant.

## Extensive intermolecular RNA–RNA interactions are required for formation of dynamic mesh-like condensates

Next, we set out to address how RNAs with LDRs induce networks. We had observed that RNAs that are unable to induce networks are predicted to form strong local structures, meaning that they have a high propensity for intramolecular interactions (*Figure 5A*). This led us to hypothesize that network formation is caused by intermolecular RNA–RNA interactions mediated by the LDRs of mRNAs (*Figure 5B*). To test this, we performed native gel electrophoresis with sphere-forming and

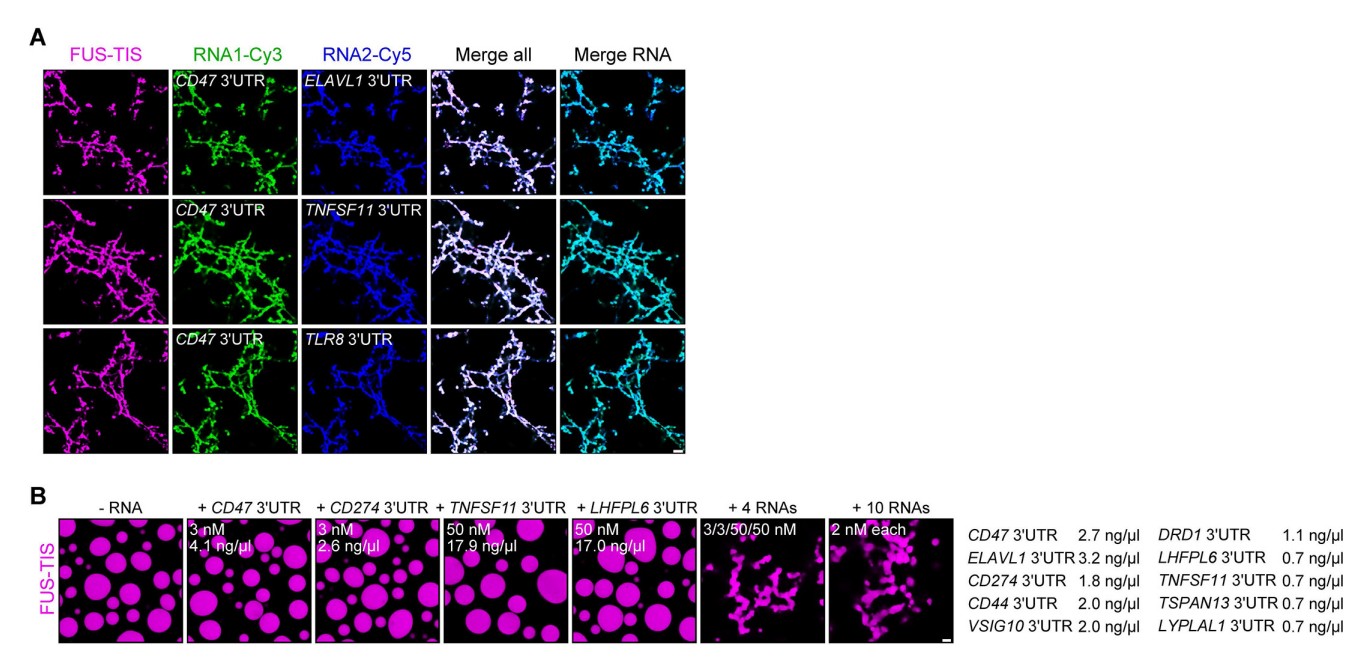

**Figure 4.** RNAs work additively to induce formation of granule networks in vitro. (**A**) RNAs co-localize in mesh-like condensates. Representative confocal images of phase separation experiments using purified mGFP-FUS-TIS (10 μM) in the presence of two different in vitro transcribed RNAs that were labeled with Cy3 or Cy5 fluorescent dye, respectively. Images were taken after 16 hr of incubation. Scale bar, 5 μm. (**B**) Representative confocal images of phase separation experiments using purified mGFP-FUS-TIS (10 μM) in the presence of a single network-forming RNA at suboptimal concentration or in the presence of 4 or 10 network-forming RNAs, each at suboptimal concentration. Images were taken after 16 hr of incubation. Scale bar, 2 μm.

network-forming RNAs. We observed the appearance of diverse RNA species with high molecular weight only with the network-forming RNAs (*Figure 5C*). The observed smear is not due to degradation as the denaturing gel demonstrates that the used RNAs are intact (*Figure 5C*). This suggested that mRNAs with LDRs form higher-order RNA interactions in vitro.

To investigate if intermolecular RNA–RNA interactions are indeed the cause of mesh-like condensates, we performed in vitro reconstitution experiments with RNAs that were designed to form multivalent RNA–RNA interactions (*Figure 5D*). We selected two 3′UTRs (*TLR8* and *MYC*) that are unable to induce network formation (*Figure 2D*). However, they are able to dimerize, and this feature provides one degree of multivalency (*Figure 5D, E*, *Figure 5—figure supplement 1A*). We added two different RNA dimerization elements to their 5′ and 3′ ends to increase RNA multivalency (*Figure 5D*; *Figure 5—figure supplement 1B, C*). Adding RNA dimerization elements did not substantially change the NED values of *TLR8* and *MYC* 3′UTRs. (*Figure 5—figure supplement 1D, E*), but this strategy enables intermolecular RNA–RNA interactions and allows the formation of a complex RNA network, demonstrated by native gel electrophoresis (*Figure 5D, E*). The RNA dimerization elements were derived from tracrRNA/crRNA (D1) and from HIV (D2) (*Figure 5—figure supplement 1B, C*; *Skripkin et al., 1994*; *Jinek et al., 2012*; *Paillart et al., 2004*; *Khan et al., 2019*).

A phase separation experiment with FUS-TIS confirmed that the addition of the two predominantly structured 3′UTRs (*TLR8* and *MYC*) generates sphere-like condensates, whereas the addition of RNAs capable of forming a crosslinked RNA network (D1-TLR8-D2 and D1-MYC-D2), which we call an RNA matrix, induces formation of mesh-like FUS-TIS condensates (*Figure 5F*). Taken together, this in vitro reconstitution experiment demonstrated that an extensive, multivalent RNA interaction network can drive the formation of mesh-like condensates.

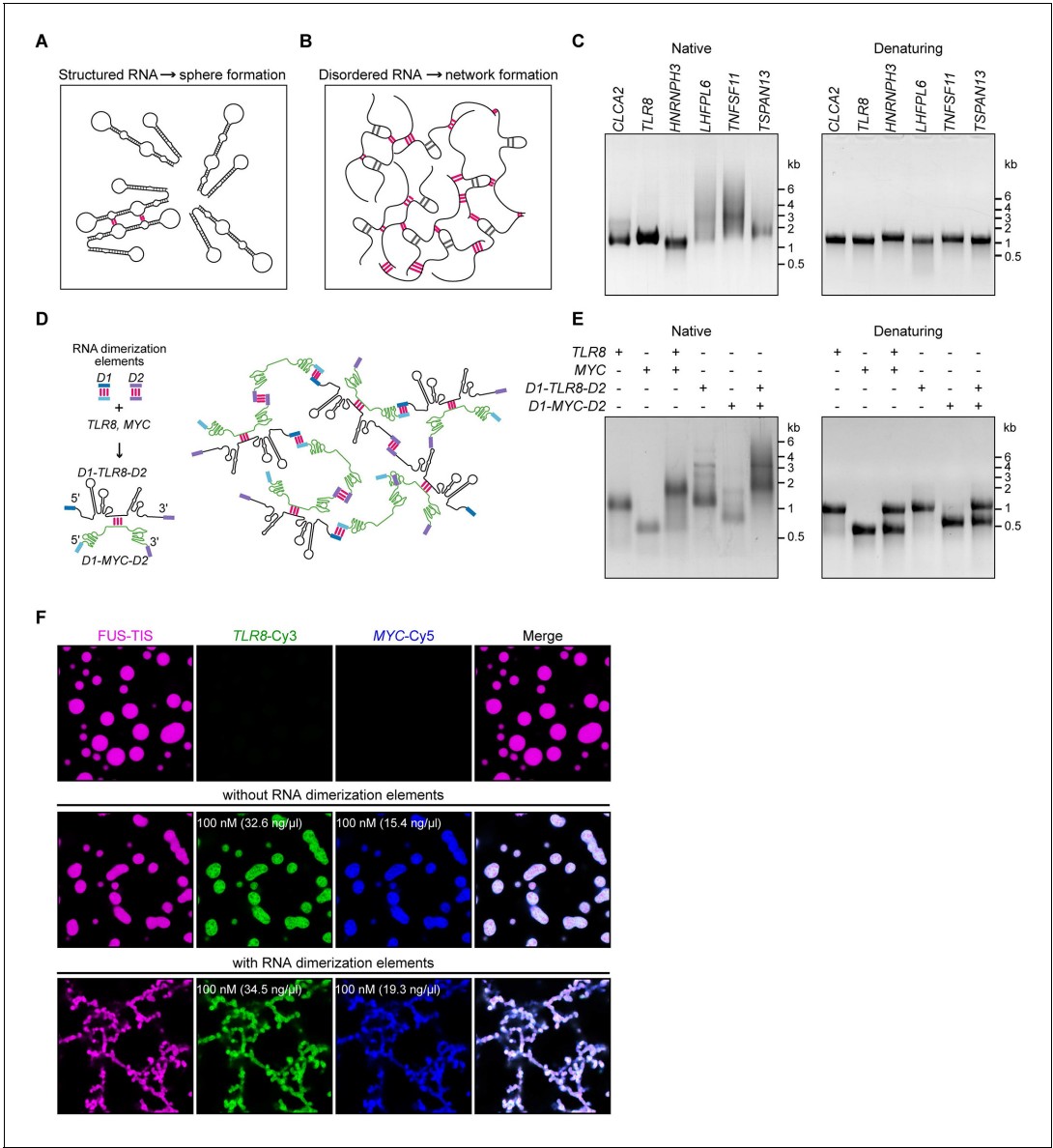

**Figure 5.** A multivalent RNA matrix is responsible for mesh-like condensate formation in vitro. (**A**) Schematic of RNAs with strong local secondary structures that are predicted to induce spherical condensates. (**B**) Schematic of RNAs with large disordered regions that form extensive intermolecular RNA–RNA interactions that are predicted to form network-like condensates. (**C**) Native and denaturing agarose gel electrophoresis of sphere-forming (lanes 1–3) and network-forming (lanes 4–6) RNAs (5 µM, each [1.7, 1.6, 1.9, 1.7, 1.8, 1.7 µg/µl]). (**D**) Schematic of a complex RNA network characterized by extensive intermolecular RNA–RNA interactions mediated by two dimerization elements (D1 and D2) that were added to structured RNAs. (**E**) Native and denaturing agarose gel electrophoresis of the indicated RNAs (1 µM, each [326, 154, 345, 193 ng/µl]). (**F**) Representative images of phase separation experiments using purified mGFP-FUS-TIS (10 µM) in the presence of the indicated Cy3- or Cy5-labeled RNAs generated by in vitro transcription after 16 hr of incubation. Scale bar, 2 µm.

The online version of this article includes the following figure supplement(s) for figure 5:

**Figure supplement 1.** Extensive intermolecular RNA–RNA interactions are responsible for formation of mesh-like condensates in vitro.

## Formation of a crosslinked mRNA network is sufficient for the reconstitution of mesh-like condensates in vivo

To investigate if mesh-like condensates can also be reconstituted in vivo, we used the same RNAs (*TLR8* and *MYC* 3'UTRs with and without dimerization elements) together with SUMO-SIM as protein component. SUMO-SIM is especially suitable for this approach as all SUMO-SIM condensates are sphere-like. Our goal was to recruit mRNAs that are able to form a pervasive RNA interaction

network into SUMO-SIM condensates using the MS2 system (*Bertrand et al., 1998*; *Berkovits and Mayr, 2015*) as this should turn the sphere-like condensates into mesh-like condensates (*Figure 6A*).

We fused SUMO-SIM to the MS2 coat protein, which can be considered as a selective RBD. The MS2 coat protein binds to RNA stem loops that were introduced in both *TLR8* and *MYC* 3'UTRs. This system allows phase separation as SUMO-SIM is a multivalent protein and at the same time recruits the two structured 3'UTRs into the condensate (*Figure 6B*). In the presence of all the required elements (SUMO-SIM fused to MS2 coat protein, *TLR8* and *MYC* 3'UTRs fused to MS2-binding sites, and the presence of the dimerization elements *D1* and *D2* in the RNAs), we observed mesh-like condensate formation in vivo, indicating that we are able to reconstitute mesh-like condensates in living cells (*Figure 6A*). Omission of MS2-binding sites or the MS2 coat protein prevents

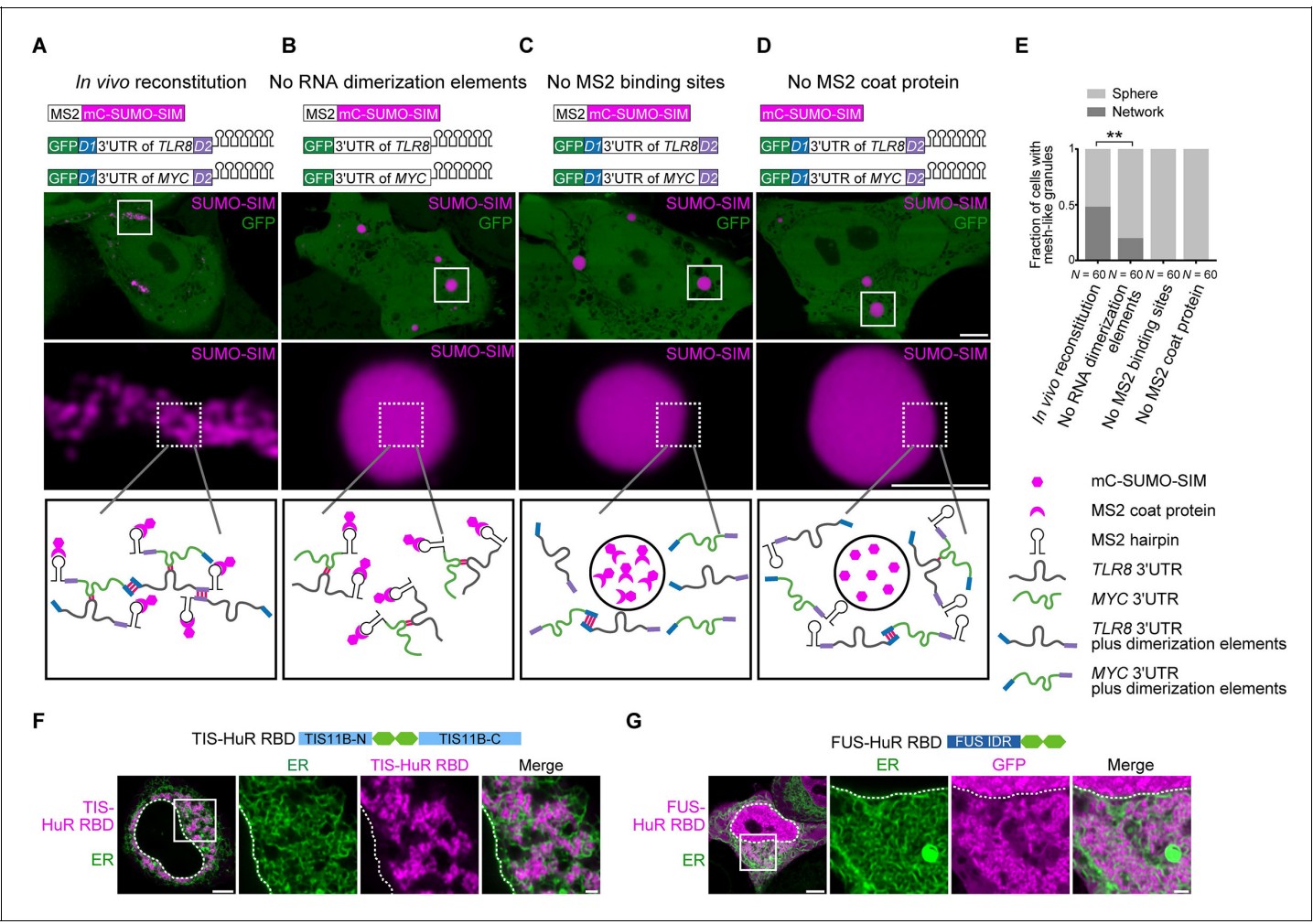

**Figure 6.** Formation of an extensive mRNA network is sufficient for the reconstitution of mesh-like condensates in vivo. (A) Representative confocal image of in vivo reconstitution of mesh-like condensates using the MS2 system. mCherry-SUMO-SIM fused to the MS2 coat protein was transfected into HeLa cells. Constructs containing eGFP-fused 3'UTRs of *TLR8* and *MYC* with MS2-binding sites and with the RNA dimerization elements *D1* and *D2* were co-transfected. Bottom: higher magnification of the indicated regions. Scale bars, 5 μm (overview) and 2 μm (zoom-in). Images were taken 16 hr after transfection. mC: mCherry. (B-D) As in (A), but one of the indicated components was omitted. (E) Quantification of the fraction of cells with mesh-like granules in the conditions shown in (A–D). Mann–Whitney test: Z = −3.5, **p=0.001. (F) Confocal live-cell imaging of HeLa cells after the transfection of mCherry-tagged TIS-HuR chimera, containing the RRM1/2 of HuR as well as the N- and C-terminal regions of TIS11B. All granules are mesh-like. Scale bars, 5 μm (overview) and 1 μm (zoom-in). (G) Same as (F), but after transfection of mGFP-tagged FUS-HuR chimera, containing the RRM1/2 of HuR as well as the FUS IDR. All granules are mesh-like.

The online version of this article includes the following source data and figure supplement(s) for figure 6:

**Source data 1.** Transcriptome-wide analysis on normalized ensemble diversity values of 3'UTRs.

**Figure supplement 1.** mRNAs with large disordered regions are enriched in AU-rich elements.

recruitment of the mRNAs to the multivalent protein and completely prevents the formation of mesh-like condensates, indicating that RNA is absolutely required for mesh-like condensate formation (*Figure 6C, D*). Omission of the two RNA dimerization elements from the system still allows recruitment of the RNAs to the multivalent protein. As the RNAs are able to dimerize, they generate mesh-like condensates in a few cases, but form sphere-like condensates in the majority of cases (*Figure 6B, E*). However, the presence of RNA dimerization elements that allow the generation of an extensive RNA interaction network that interacts with a multivalent protein substantially increases formation of mesh-like condensates in living cells (*Figure 6A, E*). These in vivo reconstitution experiments confirm that phase separation of an RNA-binding protein, together with mostly structured RNAs, predominantly generates sphere-like condensates, whereas phase separation of the same RNA-binding protein bound to RNAs with an ability to generate an extensive interaction network form mesh-like condensates in living cells.

## Various chimeric proteins induce formation of mesh-like condensates in cells

In the in vivo reconstitution experiments, we did not use the TIS11B RBD, but we functionally replaced it by the recruitment of RNA matrix-forming RNAs. This suggests that other RBDs may also be able to generate mesh-like condensates in cells. TIS11B binds to AU-rich elements (*Peng et al., 1998*). Transcriptome-wide analyses showed that mRNAs with several AU-rich elements in their 3′UTRs have longer 3′UTRs and higher NED values (*Figure 6—figure supplement 1A, B*, *Figure 6— source data 1*). HuR is an RNA-binding protein that also binds to U- or AU-rich elements (*Lebedeva et al., 2011*; *Mukherjee et al., 2011*; *Uren et al., 2011*). Analyzing HuR PAR-CLIP data showed that HuR targets also have longer 3′UTRs and higher NED values (*Figure 6—figure supplement 1C, D*). Therefore, we examined if the HuR RBD is capable of forming mesh-like condensates in cells. In the context of two multivalent domains (TIS11B N/C-terminus or FUS-IDR), the RRM1/2 of HuR was sufficient for the generation of mesh-like condensates (*Figure 6F, G*, *Figure 6—figure supplement 1E*). Taken together, we showed that several chimeric proteins that consist of multivalent domains that are paired with RBDs that bind to RNA matrix-forming RNAs can form mesh-like condensates in cells (*Table 1*).

## An RNA matrix prevents complete fusion of condensates

To start to get at the mechanism of mesh-like condensate formation, we imaged the early phases of condensate fusion with sphere- and network-forming mRNAs. As expected, the condensates that

**Table 1.** Chimeric proteins investigated for mesh-like condensate formation.
RBDmut, RNA-binding domain mutant.

| Multivalent domain | RNA-binding domain | RNA | Diffusive pattern | Sphere-like condensate | Mesh-like condensate |
|---|---|---|---|---|---|
| | TIS11B | | √ | | |
| | HuR | | √ | | |
| FUS IDR | | | √ | | |
| SUMO-SIM | | | | √ | |
| TIS11B-N/C | TIS11B RBDmut | | | √ | |
| TIS11B-N/C | TIS11B | | | | √ |
| TIS11B-N/C | HuR | | | | √ |
| FUS IDR | TIS11B | | | | √ |
| FUS IDR | HuR | | | | √ |
| SUMO-SIM | TIS11B | | | | √ |
| SUMO-SIM | MS2 | | | √ | |
| SUMO-SIM | MS2 | Singlevalent RNA | | √ | |
| SUMO-SIM | MS2 | Multivalent RNA | | | √ |

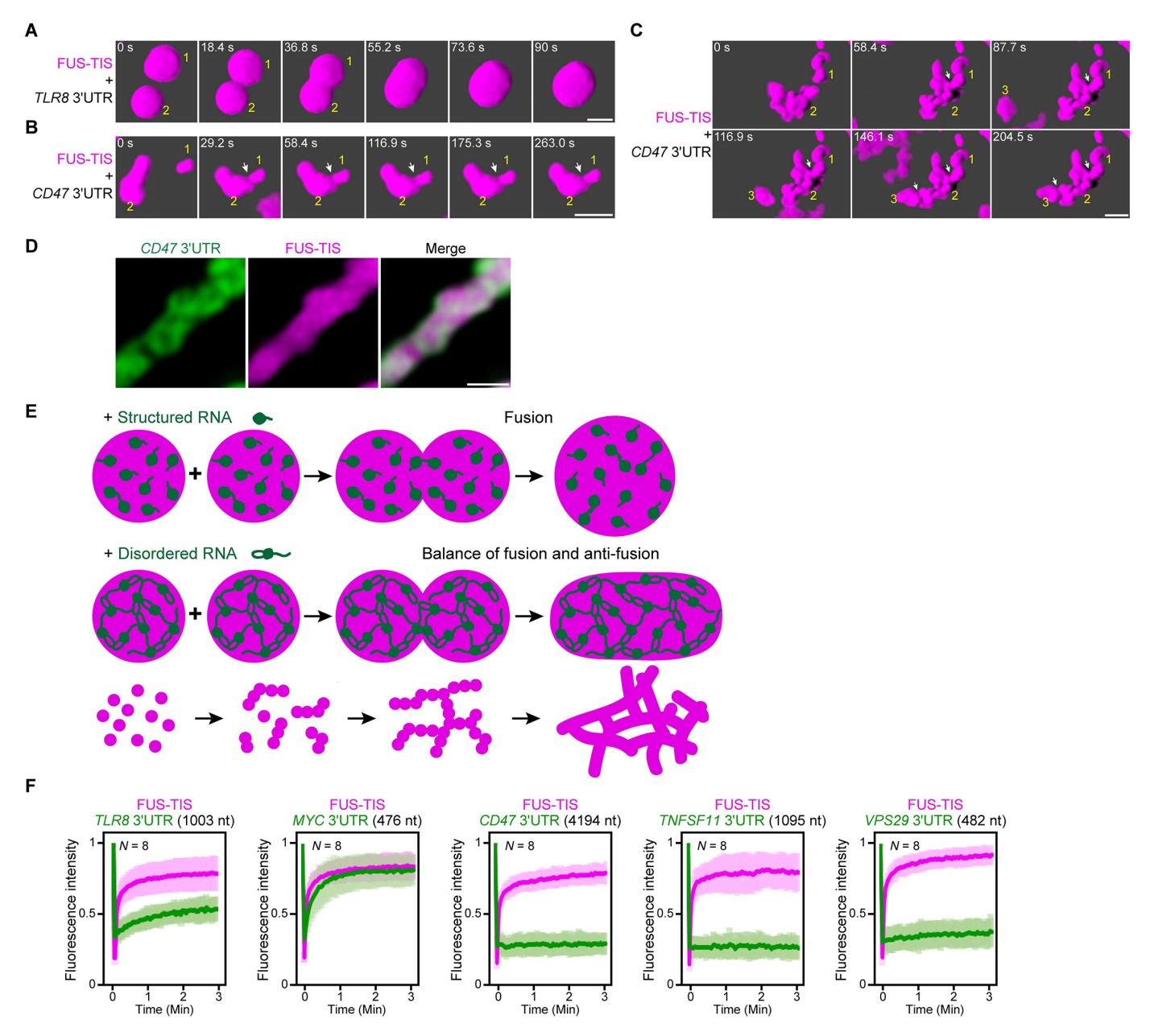

**Figure 7.** An RNA matrix prevents full fusion of spherical condensates, thus promoting arrangement into filamentous structures in vitro. (**A**) Confocal 3D time-lapse imaging of phase separation experiments using purified mGFP-FUS-TIS (10 μM) in the presence of the *TLR8* 3′UTR (200 nM) after 30 min of incubation. Scale bar, 2 μm. Snapshots show a fusion event of two FUS-TIS condensates. (**B**) Same as (**A**), but in the presence of the *CD47* 3′UTR (30 nM). Snapshots show the contact of two FUS-TIS condensates. As they do not fully mix, they grow into condensates with irregular shapes. The contact site is indicated by the white arrow. (**C**) Same as (**B**). Snapshots show two fusion events between FUS-TIS condensates 1 and 2 and between 2 and 3 and demonstrate how irregularly shaped condensates grow into large filamentous networks. (**D**) Representative high-resolution confocal images of phase separation experiments using purified mGFP-FUS-TIS (10 μM) in the presence of Cy5-labeled *CD47* 3′UTR RNA after 16 hr of incubation. Scale bar, 1 μm. (**E**) Model showing how structured RNAs induce spherical condensates and how RNAs with large unstructured regions induce formation of filamentous and mesh-like condensates. Purple indicates the condensate. For details, see text. (**F**) Fluorescence recovery after photobleaching of mGFP-FUS-TIS and the indicated Cy5-labeled RNAs performed at 2 hr after setting up the phase separation experiments.

The online version of this article includes the following figure supplement(s) for figure 7:

**Figure supplement 1.** Extensive intermolecular RNA–RNA interactions are responsible for formation of mesh-like condensates in vitro.

contain the predominantly structured 3'UTR of *TLR8* fully fuse within 90 s (*Figure 7A*). In contrast, when two condensates that contain RNAs with LDRs, such as the 3'UTR of *CD47*, come into contact, they largely retain their shape and do not fully mix (*Figure 7B*). However, the two condensates connect at the contact sites; this enables linkage of individual condensates and allows them to grow into large assemblies with irregular shapes (*Figure 7C*). It is worth pointing out that despite the filamentous condensate shape the protein components are still highly mobile as was shown by FRAP experiments performed at 2 and 16 hr after mixing (*Figure 2B*, *Figure 2—figure supplement 1F*).

To get a better understanding of the morphology of the mRNAs within sphere- and mesh-like condensates, we performed high-resolution imaging. We observed that the distribution of sphere-forming mRNAs is relatively uniform in the FUS-TIS condensates (*Figure 7—figure supplement 1A*). In contrast, the network-forming mRNAs are unevenly distributed and generate an underlying scaffold for the phase-separated protein that resembles a skeleton (*Figure 7D*).

We propose the following speculative model (*Figure 7E*). Fusion of condensates that contain structured RNAs allows the mixing of components, thus resulting in the formation of larger spheres. Highly structured RNAs can be viewed to be globule-like, and their movement within condensates is only somewhat restricted due to very weak interactions (*Hyman et al., 2014*; *Ranganathan and Shakhnovich, 2020*). In contrast, the LDRs within RNAs (depicted as loops and tails) form pervasive interactions (*Figure 5E, F*), thus forming an RNA matrix. When two condensates that contain such an RNA skeleton come into contact, the semi-rigidity of the crosslinked RNA network prevents full fusion and only allows mixing of the components at the contact sites. This arranges the condensates as beads-on-a-string and results in formation of filamentous mesh-like condensates whose protein components are dynamic. These data suggest that the fusion force provided by the surface tension of liquid condensates is counteracted by an anti-fusion effect provided by the underlying RNA matrix, thus forming a mesh-like condensate.

This model predicts that sphere-forming mRNAs are mobile and exchange with their environment, whereas network-forming mRNAs are static within the mesh-like condensates. We used FRAP on several RNAs to test the prediction (*Figure 7F*, *Figure 7—figure supplement 1B*). We observed indeed fast fluorescence recovery of sphere-forming mRNAs, including *TLR8* and *MYC*, suggesting that they are highly mobile within the condensates (*Figure 7F*, *Figure 7—figure supplement 1B*). In contrast, network-forming mRNAs of various lengths showed no fluorescence recovery (*Figure 7F*, *Figure 7—figure supplement 1B*), which supports the notion that these mRNAs act as skeleton or underlying RNA matrix for mesh-like condensates.

## Discussion

Recent studies observed biomolecular condensates in cells that have a mesh-like morphology. These condensates include TIS granules and L-bodies (*Ma and Mayr, 2018*; *Neil et al., 2020*). In the process of investigating the morphology of TIS granules, we generated chimeric proteins with multivalent domains fused to the RBD of TIS11B and observed that these proteins generated mesh-like condensates with highly dynamic protein components in cells. This observation seemed paradoxical at first glance as surface tension is known to promote sphere formation of liquid-like droplets (*Hyman et al., 2014*). This raised the question of how can liquid-like condensates have shapes other than spheres? To address this question, we used a reductionist approach and reconstituted mesh-like condensates in vitro and in vivo. We found that multivalent RNA–RNA interactions drive the formation of mesh-like condensates.

Our in vivo reconstitution of mesh-like condensates revealed that the minimal components consist of a multivalent RNA-binding protein that recruits RNAs that are able to form extensive intermolecular mRNA–mRNA interactions (*Figure 6A*). In contrast, if the same multivalent RNA-binding protein recruits RNAs without pervasive intermolecular interactions – that were generated through omission of RNA dimerization elements – mostly sphere-like condensates are generated in cells (*Figure 6B*). In the in vitro experiments, both sphere-forming and network-forming RNAs are equally recruited to the condensates (*Figure 2A, D*) as the chosen RNAs had comparable numbers of AU-rich elements, the known binding sites for the TIS11B RBD. This indicates that both mesh-like and sphere-like condensates contain RNAs, but they contain different kinds of RNAs. This is further exemplified by the FRAP experiments that revealed that RNAs that form sphere-like condensates show dynamic behavior, whereas RNAs that form mesh-like condensates have little or no FRAP recovery (*Figure 7F*). The

non-dynamic nature of these RNAs supports our model that pervasive interactions of these RNAs form an underlying RNA matrix that causes the mesh-like condensate morphology. Importantly, the characteristic FRAP behavior with dynamic protein components and non-dynamic RNA components was not only observed in our in vitro reconstitution experiments of mesh-like condensates, but also in the naturally occurring mesh-like L-bodies in frog oocytes (*Neil et al., 2020*).

Our results further indicate that a large fraction of 3′UTRs has high RNA multivalency. Our results show that the extent of RNA multivalency can be tested experimentally through native RNA gel analysis (*Figure 5C, E*). The multivalent 3′UTRs have a high propensity to form pervasive intermolecular interactions. When they are present at high concentration, they form an RNA skeleton for mesh-like condensates. The multivalent 3′UTRs are further predicted to adopt diverse structural conformations (*Figure 3*). As this feature correlates with 'unstructured-ness' of the RNA, our observations suggest that the formation of a multivalent RNA interaction network is an emergent property of unstructured mRNA. The most important RNA feature to induce formation of mesh-like condensates is their capacity for pervasive intermolecular RNA–RNA interactions. This means that predominantly structured RNAs that are somehow able to form extensive intermolecular interactions are also able to form mesh-like condensates. This was revealed by our in vivo reconstitution experiment where known RNA dimerization motifs that are mostly structured but that form a pervasive RNA matrix through multivalent RNA–RNA interactions are sufficient for mesh-like condensate formation in the presence of a multivalent RNA-binding protein (*Figure 6*).

It remains to be shown if pervasive RNA–RNA interactions are the basis for the mesh-like morphology of TIS granules. It is possible that binding of the TIS11B RBD to its target RNA could induce a conformational change in TIS11B, thus inducing condensation and branching. Such a mechanism was recently shown to be the case for G3BP1 that forms sphere-like granules (*Yang et al., 2020*; *Sanders et al., 2020*; *Guillén-Boixet et al., 2020*). Alternatively, the RNAs that are bound by TIS11B may have a high propensity for intermolecular RNA–RNA interactions, thus creating mesh-like TIS granules. This model is supported by our transcriptome-wide analysis on NED values. We found that the target mRNAs of TIS11B – which are characterized by the presence of multiple AU-rich elements in their 3′UTRs – have higher NED values than 3′UTRs without AU-rich elements (*Figure 6—figure supplement 1A, B*). This predicts that mRNAs that are enriched in TIS granules are less structured and have a higher tendency to form multivalent RNA–RNA interactions, thus making it likely that an underlying RNA matrix causes the mesh-like morphology of TIS granules.

## 3′UTRs may fulfill structural roles in the cytoplasm

Our data suggest that, in addition to acting as information templates for protein synthesis, mRNAs and, in particular, 3′UTRs may further fulfill roles as structural elements for cytoplasmic RNA granules. This is reminiscent of the scaffolding roles of lncRNAs in nuclear bodies, including paraspeckles (*Chujo et al., 2016*). Both 3′UTRs and lncRNAs have a similar AU-content and hexamer composition that differs substantially from the coding region. This finding provides further support for the scaffolding role of 3′UTRs because a higher AU-content is associated with decreased RNA structure and higher RNA flexibility (*Niazi and Valadkhan, 2012*). Here, we identified two RBDs (TIS11B and HuR) that both bind to longer 3′UTRs with high NED values that are able to form mesh-like condensates in cells. Therefore, it is likely that interacting 3′UTRs do not only act as RNA matrix for TIS granules and L-bodies, but may also scaffold additional cytoplasmic membraneless compartments, possibly on other membrane surfaces or on the cytoskeleton (*Smith et al., 2020*; *Béthune et al., 2019*).

Formation of filamentous and mesh-like condensates is not restricted to the chimeric RNA-binding proteins used here but was also observed by others (*Lin et al., 2015*; *Boeynaems et al., 2019*; *Smith et al., 2020*; *Neil et al., 2020*). For example, the mixing of a PR30 peptide with two homopolymeric RNA species with perfect base-pairing capabilities (poly-rA plus poly-rU) induced formation of filamentous condensates (*Boeynaems et al., 2019*). The PR30 condensates showed little fluorescent recovery upon FRAP, thus, suggesting that physical cross-linking of the base-paired RNA structures and the PR molecules arrests phase separation (*Boeynaems et al., 2019*). It is likely that this represents a pathological feature of PR molecules that were shown previously to reduce FRAP recovery in many phase separation systems (*Lee et al., 2016*). In contrast, in our system, we used the TIS11B and HuR RBDs that bind to 3′UTRs that contain mixtures of structured and unstructured regions under physiological conditions. These observations indicate that some filamentous condensates are solid, whereas others are dynamic, and their material properties are regulated by the

extent and strength of RNA–protein and RNA–RNA interactions (*Ferrandon et al., 1997*; *Jambor et al., 2011*; *Trcek et al., 2015*; *Van Treeck et al., 2018*; *Trcek et al., 2020*).

## The large surface area of mesh-like condensates supports reactions on the surface

Although the physiological relevance of mesh-like condensates is currently largely unknown, our study indicates that mesh-like condensates cannot simply be viewed as aggregation. Surface tension enforces a low surface to volume ratio for liquid-like spheres (*Hyman et al., 2014*). In contrast, mesh-like condensates have a larger surface to volume ratio. This gives them an advantage for reactions that occur on their surface. As TIS granules are intertwined with the ER, they share a large interface (*Ma and Mayr, 2018*). Our previous data suggests that the mesh-like shape of the TIS granule network is necessary for its function during translation of membrane proteins. We showed previously that translation in the TIGER domain enables protein complex formation of membrane proteins and promotes their trafficking to the plasma membrane (*Ma and Mayr, 2018*).

## Intermolecular mRNA interactions and unstructured mRNA regions may have additional biological relevance

These observations may indicate that the mesh-like shape of TIS granules is especially important for reactions that occur in the interface with the ER. However, also mRNAs that encode non-membrane proteins are translated in TIS granules. As translation of these proteins does not require them to be incorporated into membranes, it seems that for these reactions the shape of TIS granules is irrelevant. Rather, these reactions may take advantage of the local environment generated by TIS granules where a connected membraneless compartment may enable the enriched protein components to exchange relatively freely, thus allowing the sharing of RNA-binding proteins and chaperones (*Ma and Mayr, 2018*).

One potential biological implication for intermolecular mRNA–mRNA interactions in TIS granules could be to facilitate co-translational protein complex assembly. Although the majority of protein complexes form co-translationally in yeast (*Shiber et al., 2018*), it is unclear how the protein subunits come into proximity. It is possible that the physical interaction between LDRs in 3′UTRs provides the necessary proximity of two translating ribosomes and their nascent peptide chains to promote complex assembly (*Shiber et al., 2018*; *Mayr, 2018*).

## The morphogenesis of organelle networks is driven by the interplay of two opposing forces

We noticed that the generation of mesh-like networks – either lipid membrane-enclosed or membraneless – is conceptually similar. Our model proposes that mesh-like condensates with dynamic protein components are generated from the interplay of two forces (*Figure 7E*). One force promotes fusion and is driven by the surface tension of the liquid condensates. The other force prevents full fusion and is provided by the underlying RNA matrix that forms a semirigid skeleton and only allows fusion of the condensates at the contact sites. Such interplay of two antagonistic forces was also observed upon in vitro reconstitution of the membrane-enclosed tubular ER network (*Powers et al., 2017*). Sphere-like liposomes were mixed with proteins that exert two opposing forces: One of the proteins promotes liposome fusion and network assembly, whereas the other promotes fragmentation and disassembly. Taken together, these data indicate that membrane-enclosed and membraneless network structures result from the balance of two opposing forces.

RNA is probably the most versatile molecule in cells (*Mayr, 2017*). Through RNA mimicry, viral RNAs can mimic the shape of cellular tRNAs (*Colussi et al., 2014*). RNA can also mimic the shape of proteins and protein interaction surfaces (*Athanassiou et al., 2004*; *Shao and Hegde, 2016*; *Mizrak and Morgan, 2019*). Here, we found that the network structure and morphogenesis of the tubular ER that is generated by proteins and lipids can be mimicked by phase-separated condensates formed by protein and RNA. Our work showed that dynamic subcellular structures with complex shapes can be generated through phase separation without the need for lipid membranes.

# Materials and methods

## Key resources table

| Reagent type (species) or resource | Designation | Source or reference | Identifiers | Additional information |
|---|---|---|---|---|
| Cell line (*Homo sapiens*) | HeLa | Jonathan S. Weissman | N/A | A human cervical cancer cell line (female origin). |
| Strain, strain background (*Escherichia coli*) | BL21(DE3) | NEB | C2527H | Chemically competent *E. coli* cells. |
| Antibody | Anti-α-tubulin (mouse monoclonal) | Sigma-Aldrich | Cat# T9026, RRID:AB_477593 | WB (1:5000). |
| Antibody | Anti-mCherry (mouse monoclonal) | Abcam | Cat# ab125096, RRID:AB_11133266 | WB (1:5000). |
| Antibody | Anti-HuR (rabbit polyclonal) | Millipore | Cat# 07-1735, RRID:AB_1977173 | WB (1:2000). |
| Antibody | IRDye 680RD anti-rabbit IgG secondary antibody (donkey polyclonal) | LI-COR Biosciences | Cat# 926-68073, RRID:AB_10954442 | WB (1:10,000). |
| Antibody | IRDye 800CW anti-mouse IgG secondary antibody (donkey polyclonal) | LI-COR Biosciences | Cat# 926–32212, RRID:AB_621847 | WB (1:10,000). |
| Transfected construct (human) | pcDNA-SP-GFP-CD47-LU | *Berkovits and Mayr, 2015* | N/A | See Materials and methods. |
| Transfected construct (human) | pcDNA-GFP-ELAVL1-LU | *Ma and Mayr, 2018* | N/A | See Materials and methods. |
| Transfected construct (human) | pcDNA-SP-GFP-CD274-UTR | *Ma and Mayr, 2018* | N/A | See Materials and methods. |
| Transfected construct (human) | pcDNA-SP-GFP-FUS-UTR | *Ma and Mayr, 2018* | N/A | See Materials and methods. |
| Transfected construct (human) | pcDNA-GFP-SEC61B | *Ma and Mayr, 2018* | N/A | See Materials and methods. |
| Transfected construct (human) | pcDNA-mCherry-SEC61B | *Ma and Mayr, 2018* | N/A | See Materials and methods. |
| Transfected construct (human) | pcDNA-mCherry-TIS11B | *Ma and Mayr, 2018* | N/A | See Materials and methods. |
| Transfected construct (human) | pcDNA-mCherry-TIS11B CC | This paper | N/A | See Materials and methods. |
| Transfected construct (human) | pcDNA-mCherry-TIS11B FF | This paper | N/A | See Materials and methods. |
| Transfected construct (human) | pcDNA-mCherry-TIS11B KK | This paper | N/A | See Materials and methods. |
| Transfected construct (human) | pcDNA-mCherry-TIS11B RK | This paper | N/A | See Materials and methods. |
| Transfected construct (human) | pcDNA-mCherry-TIS-HuR RBD | This paper | N/A | See Materials and methods. |
| Transfected construct (human) | pmCherry-SUMO10-SIM5 | Liam J. Holt (NYU) | N/A | See Materials and methods. |
| Transfected construct (human) | pmCherry-SUMO10-SIM5-TIS | This paper | N/A | See Materials and methods |

*Continued on next page*

*Continued*

| Reagent type (species) or resource | Designation | Source or reference | Identifiers | Additional information |
|---|---|---|---|---|
| Transfected construct (human) | pcDNA-mGFP-FUS-TIS | This paper | N/A | See Materials and methods. |
| Recombinant DNA reagent | pET28a | Dirk Remus (MSKCC) | N/A | Bacterial expression vector. |
| Recombinant DNA reagent | pDZ2087 | Addgene | Cat# 92414 | Bacterial expression of TEV protease. |
| Recombinant DNA reagent | pET28a-6xHis-MBP-mGFP-FUS-TIS-Strep-Tag II | This paper | N/A | Bacterial expression of 6xHis-MBP-mGFP-FUS-TIS-Strep Tag II. See Materials and methods. |
| Recombinant DNA reagent | T7-*CD47* 3′UTR | This paper | N/A | T7 RNA polymerase-based in vitro transcription. See Materials and methods. |
| Recombinant DNA reagent | T7-*ELAVL1* 3′UTR | This paper | N/A | T7 RNA polymerase-based in vitro transcription. See Materials and methods. |
| Recombinant DNA reagent | T7-*CD274* 3′UTR | This paper | N/A | T7 RNA polymerase-based in vitro transcription. See Materials and methods. |
| Recombinant DNA reagent | T7-*FUS* 3′UTR | This paper | N/A | T7 RNA polymerase-based in vitro transcription. See Materials and methods. |
| Recombinant DNA reagent | T7-*CD44* 3′UTR | This paper | N/A | T7 RNA polymerase-based in vitro transcription. See Materials and methods. |
| Recombinant DNA reagent | T7-*VSIG10* 3′UTR | This paper | N/A | T7 RNA polymerase-based in vitro transcription. See Materials and methods. |
| Recombinant DNA reagent | T7-*IL10* 3′UTR | This paper | N/A | T7 RNA polymerase-based in vitro transcription. See Materials and methods. |
| Recombinant DNA reagent | T7-*TNFSF11* 3′UTR | This paper | N/A | T7 RNA polymerase-based in vitro transcription. See Materials and methods. |
| Recombinant DNA reagent | T7-*GPR39* 3′UTR | This paper | N/A | T7 RNA polymerase-based in vitro transcription. See Materials and methods. |
| Recombinant DNA reagent | T7-*TLR8* 3′UTR | This paper | N/A | T7 RNA polymerase-based in vitro transcription. See Materials and methods. |
| Recombinant DNA reagent | T7-*GPR34* 3′UTR | This paper | N/A | T7 RNA polymerase-based in vitro transcription. See Materials and methods. |
| Recombinant DNA reagent | T7-*TNFAIP6* 3′UTR | This paper | N/A | T7 RNA polymerase-based in vitro transcription. See Materials and methods. |
| Recombinant DNA reagent | T7-*MYC* 3′UTR | This paper | N/A | T7 RNA polymerase-based in vitro transcription. See Materials and methods. |

*Continued on next page*

*Continued*

| Reagent type (species) or resource | Designation | Source or reference | Identifiers | Additional information |
|---|---|---|---|---|
| Recombinant DNA reagent | T7-*PLA2G4A* 3′UTR | This paper | N/A | T7 RNA polymerase-based in vitro transcription. See Materials and methods. |
| Recombinant DNA reagent | T7-*HEATR5B* 3′UTR | This paper | N/A | T7 RNA polymerase-based in vitro transcription. See Materials and methods. |
| Recombinant DNA reagent | T7-*PPP1R3F* 3′UTR | This paper | N/A | T7 RNA polymerase-based in vitro transcription. See Materials and methods. |
| Recombinant DNA reagent | T7-*DRD1* 3′UTR | This paper | N/A | T7 RNA polymerase-based in vitro transcription. See Materials and methods. |
| Recombinant DNA reagent | T7-*FAM72B* 3′UTR | This paper | N/A | T7 RNA polymerase-based in vitro transcription. See Materials and methods. |
| Recombinant DNA reagent | T7-*MCOLN2* 3′UTR | This paper | N/A | T7 RNA polymerase-based in vitro transcription. See Materials and methods. |
| Recombinant DNA reagent | T7-*TSPAN13* 3′UTR | This paper | N/A | T7 RNA polymerase-based in vitro transcription. See Materials and methods. |
| Recombinant DNA reagent | T7-*LHFPL6* 3′UTR | This paper | N/A | T7 RNA polymerase-based in vitro transcription. See Materials and methods. |
| Recombinant DNA reagent | T7-*FAM174A* 3′UTR | This paper | N/A | T7 RNA polymerase-based in vitro transcription. See Materials and methods. |
| Recombinant DNA reagent | T7-*VPS29* 3′UTR | This paper | N/A | T7 RNA polymerase-based in vitro transcription. See Materials and methods. |
| Recombinant DNA reagent | T7-*ADPGK* 3′UTR | This paper | N/A | T7 RNA polymerase-based in vitro transcription. See Materials and methods. |
| Recombinant DNA reagent | T7-*ASPN* 3′UTR | This paper | N/A | T7 RNA polymerase-based in vitro transcription. See Materials and methods. |
| Recombinant DNA reagent | T7-*CASP8* 3′UTR | This paper | N/A | T7 RNA polymerase-based in vitro transcription. See Materials and methods. |
| Recombinant DNA reagent | T7-*CLCA2* 3′UTR | This paper | N/A | T7 RNA polymerase-based in vitro transcription. See Materials and methods. |

*Continued on next page*

Continued

*Continued*

| Reagent type (species) or resource | Designation | Source or reference | Identifiers | Additional information |
|---|---|---|---|---|
| Recombinant DNA reagent | T7-*EOMES* 3′UTR | This paper | N/A | T7 RNA polymerase-based in vitro transcription. See Materials and methods. |
| Recombinant DNA reagent | T7-*ESCO1* 3′UTR | This paper | N/A | T7 RNA polymerase-based in vitro transcription. See Materials and methods. |
| Recombinant DNA reagent | T7-*GLYATL3* 3′UTR | This paper | N/A | T7 RNA polymerase-based in vitro transcription. See Materials and methods. |
| Recombinant DNA reagent | T7-*HNRNPH3* 3′UTR | This paper | N/A | T7 RNA polymerase-based in vitro transcription. See Materials and methods. |
| Recombinant DNA reagent | T7-*HOGA1* 3′UTR | This paper | N/A | T7 RNA polymerase-based in vitro transcription. See Materials and methods. |
| Recombinant DNA reagent | T7-LPAR4 3′UTR | This paper | N/A | T7 RNA polymerase-based in vitro transcription. See Materials and methods. |
| Recombinant DNA reagent | T7-*LRBA* 3′UTR | This paper | N/A | T7 RNA polymerase-based in vitro transcription. See Materials and methods. |
| Recombinant DNA reagent | T7-*LYPLAL1* 3′UTR | This paper | N/A | T7 RNA polymerase-based in vitro transcription. See Materials and methods. |
| Recombinant DNA reagent | T7-*ODF2* 3′UTR | This paper | N/A | T7 RNA polymerase-based in vitro transcription. See Materials and methods. |
| Recombinant DNA reagent | T7-*PRKDC* 3′UTR | This paper | N/A | T7 RNA polymerase-based in vitro transcription. See Materials and methods. |
| Recombinant DNA reagent | T7-*RHOA* 3′UTR | This paper | N/A | T7 RNA polymerase-based in vitro transcription. See Materials and methods. |
| Recombinant DNA reagent | T7-*SHQ1* 3′UTR | This paper | N/A | T7 RNA polymerase-based in vitro transcription. See Materials and methods. |
| Recombinant DNA reagent | T7-*SLC39A6* 3′UTR | This paper | N/A | T7 RNA polymerase-based in vitro transcription. See Materials and methods. |
| Recombinant DNA reagent | T7-*SLC5A9* 3′UTR | This paper | N/A | T7 RNA polymerase-based in vitro transcription. See Materials and methods. |
| Recombinant DNA reagent | T7-*SMIM3* 3′UTR | This paper | N/A | T7 RNA polymerase-based in vitro transcription. See Materials and methods. |
| Recombinant DNA reagent | T7-*SNTN* 3′UTR | This paper | N/A | T7 RNA polymerase-based in vitro transcription. See Materials and methods. |

*Continued on next page*

Continued

| Reagent type (species) or resource | Designation | Source or reference | Identifiers | Additional information |
|---|---|---|---|---|
| Recombinant DNA reagent | T7-*SOSTDC1* 3′UTR | This paper | N/A | T7 RNA polymerase-based in vitro transcription. See Materials and methods. |
| Recombinant DNA reagent | T7-*STBD1* 3′UTR | This paper | N/A | T7 RNA polymerase-based in vitro transcription. See Materials and methods. |
| Recombinant DNA reagent | T7-*TP53TG3* 3′UTR | This paper | N/A | T7 RNA polymerase-based in vitro transcription. See Materials and methods. |
| Recombinant DNA reagent | T7-*TTC17* 3′UTR | This paper | N/A | T7 RNA polymerase-based in vitro transcription. See Materials and methods. |
| Sequence-based reagent | Biotinylated RNA oligo, *TNFα* ARE-1 | *Ma and Mayr, 2018* | RNA oligonucleotides | 5′-CACUUGUG AUUAUUUAUU AUUUAUUUAUUAU UUAUUUAUUUA −3′ |
| Peptide, recombinant protein | FUS-TIS | This paper | N/A | Recombinant 6xHis-MBP-mGFP-FUS-TIS-Strep Tag II protein purified from bacteria. See Materials and methods. |
| Peptide, recombinant protein | Bovine serum albumin (BSA) | New England Biolab | Cat# B9000S | |
| Commercial assay or kit | Streptavidin C1 beads | Invitrogen | Cat# 65002 | Streptavidin pulldown assay. |
| Commercial assay or kit | QuikChange Lightning Multi Site-Directed Mutagenesis Kit | Agilent Technologies | Cat# 210513 | Site-directed mutagenesis. |
| Commercial assay or kit | MEGAscript T7 Transcription Kit | Invitrogen | Cat# AMB13345 | In vitro T7 transcription. |
| Commercial assay or kit | Quick Star Bradford Protein Assay Kit | Bio-Rad | Cat# 5000202 | Bradford assay – protein quantitation. |
| Chemical compound, drug | Lipofectamine 2000 | Invitrogen | Cat# 11668019 | |
| Chemical compound, drug | Dextran T500 | PHARMACOSMOS | Cat# 40030 | |
| Chemical compound, drug | Desthiobiotin | Sigma-Aldrich | Cat# D1411-1G | |
| Chemical compound, drug | Zinc chloride | Sigma-Aldrich | Cat# 793523-100G | |
| Chemical compound, drug | Imidazole | Sigma-Aldrich | Cat# I2399-100G | |
| Chemical compound, drug | IPTG | Gold Biotechnology | Cat# I2481-EZ10 | |

*Continued on next page*

*Continued*

| Reagent type (species) or resource | Designation | Source or reference | Identifiers | Additional information |
|---|---|---|---|---|
| Chemical compound, drug | PMSF | Sigma-Aldrich | Cat# 11359061001 | |
| Chemical compound, drug | DTT | Sigma-Aldrich | Cat# 10708984001 | |
| Software, algorithm | FIJI | NIH | https://fiji.sc/ | |
| Software, algorithm | ZEN | ZEISS | https://www.zeiss.com/microscopy/int/downloads/zen.html | |
| Software, algorithm | GraphPad Prism 7 | GraphPad Software | https://www.graphpad.com/scientific-software/prism | |
| Software, algorithm | Odyssey | LI-COR Biosciences | https://www.licor.com/bio/products/imaging_systems/odyssey/ | |
| Other | Ni-NTA Agarose | Qiagen | Cat# 30230 | His tag purification. |
| Other | StrepTrap column | GE Healthcare | Cat# 28907547 | Strep tag II purification. |
| Other | Amicon Ultra-centrifugal filters-50K | EMD Millipore | Cat# UFC905024 | Concentrating protein samples. |
| Other | 384-well glass-bottom microplate | Greiner Bio-One | Cat# M4437-16EA | Glass-bottom microplate for confocal imaging. |

## Cell lines

The human cervical cancer cell line, HeLa, was a gift from the lab of Jonathan S. Weissman (UCSF), provided by Calvin H. Jan. Cells were maintained at 37°C with 5% $CO_2$ in Dulbecco's Modified Eagle Medium containing 4500 mg/l glucose, 10% heat-inactivated fetal bovine serum, 100 U/ml penicillin, and 100 mg/ml streptomycin. The cell line has not been authenticated. The cell line is free of mycoplasma. Mycoplasma detection was performed by DAPI staining.

## Constructs

All primers are reported in *Table 1*. All PCR reactions were performed using Q5 High Fidelity DNA polymerase (NEB). The basis for all mammalian expression vectors was pc-DNA-puro described previously (*Ma and Mayr, 2018*). The following inserts were also described previously mCherry-TIS11B, mCherry-SEC61B, eGFP-SEC61B, eGFP-CD47-3′UTR, eGFP-ELAVL1-3′UTR, eGFP-FUS-3′UTR, and eGFP-CD274-3′UTR (*Ma and Mayr, 2018*).

Point mutations were generated using QuikChange Lightning Multi Site-Directed Mutagenesis Kit (Agilent Technologies, #210513) if not otherwise stated. pcDNA-puro-mGFP (monomeric GFP, A207K) was generated from pcDNA-puro-eGFP using the primer eGFP A207K. For TIS11B RBD mutants, the following primers were used: TIS11B C135H, TIS11B C173H, TIS11B F137N, TIS11B F175N, TIS11B K116L, TIS11B K154L, TIS11B R116L, and TIS11B K152L. We called the RBD mutants CC, FF, KK, and RK because the mutated amino acids are C135H/C173H, F137N/F175N, K116L/K154L, and R114L/K152L, respectively.

TIS-HuR RBD contains the N-terminus of TIS11B (TIS11B N; aa 1–113, based on uniprot ID Q07352-1) fused to RRM1/2 of HuR (aa 19–189) fused to the C-terminus of TIS11B (TIS11B C); (aa 182–338). This construct was generated using PCR amplification of three overlapping fragments. Fragment 1: TIS11B N was PCR-amplified from the mCherry-TIS11B construct with primers TIS-HuR 1F and TIS-HuR 1R. Fragment 2: RRM1/2 of HuR was PCR-amplified from the eGFP-HuR-3′UTR construct with primers TIS-HuR 2F and TIS-HuR 2R. Fragment 3: TIS11B C was PCR-amplified from the

mCherry-TIS11B construct with primers TIS-HuR 3F and TIS-HuR 3R. A ligation PCR was performed to ligate Fragment 1 and Fragment 2 to generate Fragment 1–2 with primers TIS-HuR 1F and TIS-HuR 2 R-2. Then Fragment 1–2 was digested with *Hin*dIII and *Apa*I; Fragment 3 was digested with *Apa*I and *Eco*RI. To generate full-length TIS-HuR chimera, Fragment 1–2 and Fragment 3 were cloned into the pcDNA3.1-puro-mCherry vector with *Hin*dIII and *Eco*RI restriction sites.

The pmCherry-SUMO10-SIM5 construct was a gift from the lab of Liam J. Holt (NYU). To generate the SUMO-SIM-TIS (TIS11B RBD fused to the N-terminus of SUMO10-SIM5) fusion protein, two overlapping fragments were PCR-amplified. Fragment 1, mCherry, was PCR-amplified from the mCherry-TIS11B construct with primers TIS-SUMO-SIM 1F and TIS-SUMO-SIM 1R. Fragment 2, the RBD (aa 114–181) of TIS11B, was PCR-amplified from the mCherry-TIS11B construct with primers TIS-SUMO-SIM 2F and TIS-SUMO-SIM 2R. A ligation PCR was performed to generate mCherry-TIS11B RBD with primers SUMO-SIM 1F and TIS-SUMO-SIM 2R. mCherry-TIS11B RBD was cloned into the SUMO10-SIM5 construct with *Age*I and *Bsr*GI restriction sites.

For the FUS-TIS (FUS IDR fused to the N-terminus of TIS11B RBD) fusion protein, two overlapping fragments were PCR-amplified. Fragment 1: the IDR (aa 1–214) of FUS was PCR-amplified from the pcDNA-puro-eGFP-FUS-3'UTR vector with primers FUS-TIS 1F and FUS-TIS 1R. Fragment 2: the RBD (aa 114–181) of TIS11B was PCR-amplified with primers FUS-TIS 2F and FUS-TIS 2R. A final ligation PCR was performed to ligate two PCR fragments to the full-length FUS-TIS chimera with primers FUS-TIS 1F and FUS-TIS 2R. The full-length FUS-TIS was cloned into pcDNA3.1-puro-mGFP vector with *Bsr*GI and *Eco*RI restriction sites.

For the pcDNA3.1-puro-BFP-FUS-TIS construct, a nuclear export signal (nes) was added upstream of the FUS IDR to increase cytoplasmic localization. It was obtained from pcDNA3.1-puro-mCherry-nes-FUS-TIS (provided by Neil Robertson, MSKCC) using *Bsr*GI and *Eco*RI restriction sites and cloned into the pcDNA3.1-puro-BFP vector with the same restriction sites.

The RBD of HuR (aa 19–189) was PCR-amplified from the eGFP-HuR-3'UTR construct with primers HuR RBD F and FUS-HuR 2R. HuR RBD was cloned into pcDNA3.1-puro-mGFP vector with *Hin*dIII and *Eco*RI restriction sites.

To generate the FUS-HuR RBD (FUS IDR fused to the N-terminus of HuR RBD) fusion protein, two overlapping fragments were PCR-amplified. Fragment 1: the IDR (aa 1–214) of FUS was PCR-amplified from the pcDNA-puro-eGFP-FUS-3'UTR vector with primers FUS-HuR 1F and FUS-HuR 1R. Fragment 2: the RBD (aa 19–189) of HuR was PCR-amplified from the eGFP-HuR-3'UTR construct with primers FUS-HuR 2F and FUS-HuR 2R. A final ligation PCR was performed to ligate two PCR fragments to obtain FUS-HuR chimera with primers FUS-HuR 1F and FUS-HuR 2R. The full-length FUS-TIS was cloned into pcDNA3.1-puro-mGFP vector with *Hin*dIII and *Eco*RI restriction sites.

Contructs for in vivo reconstitution. To generate the eGFP-3'UTR of *TLR8* construct, the *TLR8* 3'UTR was PCR-amplified from HeLa genomic DNA with primers TLR8-MS2 F and TLR8-MS2 R. The *TLR8* 3'UTR was cloned into the pcDNA-puro-eGFP vector with *Bsr*GI and *Eco*RI restriction sites.

To generate the eGFP-3'UTR of *MYC* construct, the *MYC* 3'UTR was PCR-amplified from HeLa genomic DNA with primers MYC-MS2 F and MYC-MS2 R. The *MYC* 3'UTR was cloned into the pcDNA-puro-eGFP vector with *Bsr*GI and *Eco*RI restriction sites.

To generate the eGFP-*D1*-3'UTR of *TLR8-D2* construct (*TLR8* 3'UTR with RNA dimerization elements *D1* and *D2*), the *D1-TLR8* 3'UTR-*D2* was PCR-amplified from the *TLR8* 3'UTR PCR product with primers D1D2-MS2 F1 and D1D2-MS2 R. *D1-TLR8* 3'UTR-*D2* was cloned into the pcDNA-puro-eGFP vector with *Bsr*GI and *Eco*RI restriction sites.

To generate the eGFP-*D1*-3'UTR of *MYC-D2* construct (*MYC* 3'UTR with RNA dimerization elements *D1* and *D2*), the *D1-MYC* 3'UTR-*D2* was PCR-amplified from the *MYC* 3'UTR PCR product with primers D1D2-MS2 F2 and D1D2-MS2 R. The *D1-MYC* 3'UTR-*D2* was cloned into the pcDNA-puro-eGFP vector with *Bsr*GI and *Eco*RI restriction sites.

For the eGFP-6xMS2 binding site construct, 6xMS2 binding site was cloned into pcDNA-puro-eGFP vector with *Eco*RI and *Xho*I restriction sites.

To generate the eGFP-3'UTR of *TLR8*-6xMS2 construct, the *TLR8* 3'UTR was PCR-amplified from HeLa genomic DNA with primers TLR8-MS2 F and TLR8-MS2 R. The *TLR8* 3'UTR was cloned into the pcDNA-puro-eGFP-6xMS2 vector with *Bsr*GI and *Eco*RI restriction sites.

To generate the eGFP-3'UTR of *MYC*-6xMS2 construct, the *MYC* 3'UTR was PCR-amplified from HeLa genomic DNA with primers MYC-MS2 F and MYC-MS2 R. The *MYC* 3'UTR was cloned into the pcDNA-puro-eGFP-6xMS2 vector with *Bsr*GI and *Eco*RI restriction sites.

To generate the eGFP-*D1*-3'UTR of *TLR8-D2*-6xMS2 construct (*TLR8* 3'UTR with RNA dimerization elements *D1* and *D2*), the *D1-TLR8* 3'UTR-*D2* was PCR-amplified from the *TLR8* 3'UTR PCR product with primers D1D2-MS2 F1 and D1D2-MS2 R. The *D1-TLR8* 3'UTR-*D2* was cloned into the pcDNA-puro-eGFP-6xMS2 vector with *Bsr*GI and *Eco*RI restriction sites.

To generate the eGFP-*D1*-3'UTR of *MYC-D2*-6xMS2 construct (*MYC* 3'UTR with RNA dimerization elements *D1* and *D2*), the *D1-MYC* 3'UTR-*D2* was PCR-amplified from the *MYC* 3'UTR PCR product with primers D1D2-MS2 F2 and D1D2-MS2 R. The *D1-MYC* 3'UTR-*D2* was cloned into the pcDNA-puro-eGFP-6xMS2 vector with *Bsr*GI and *Eco*RI restriction sites.

To generate the MS2-mCherry-SUMO10-SIM5 (MS2 coat protein fused to the N-terminus of mCherry-SUMO10-SIM5), MS2-mCherry was PCR-amplified from the pcDNA-MS2-mCherry-HuR vector (*Berkovits and Mayr, 2015*) with primers MS2-SUMO-SIM F and MS2-SUMO-SIM R. pmCherry-SUMO10-SIM5 vector was digested with *Age*I and *Bsr*GI to release the mCherry fragment. The MS2-mCherry was cloned into the digested pmCherry-SUMO10-SIM5 vector with *Age*I and *Bsr*GI restriction sites.

## Transfections

Lipofectamine 2000 (Invitrogen) was used for all transfections.

## RNA oligonucleotide pulldown

To examine the RNA-binding activity of TIS11B WT and TIS11B RBD mutants, RNA oligonucleotide pulldown was performed as described previously (*Ma and Mayr, 2018*). A 3'-biotinylated RNA oligonucleotide of the *TNFα* ARE-1 (AU-rich element) was purchased from Dharmacon. mCherry-tagged constructs were transfected into HeLa cells with or without 3'-biotinylated RNA oligonucleotides. Twenty-four hours after transfection, HeLa cells were lysed with 200 µl ice-cold NP-40 lysis buffer (25 mM Tris-HCl pH 7.5, 150 mM NaCl, 1% NP-40, 1 mM EDTA) for 30 min. Then, cell lysates were spun down at 20,000 *g* for 10 min at 4℃. The supernatant was transferred to a pre-cooled tube and diluted with 300 µl ice-cold dilution buffer (10 mM Tris-HCl pH 7.5, 150 mM NaCl, 0.5 mM EDTA). Streptavidin C1 beads (Invitrogen) were added to each tube and rotated for 1 hr at 4℃. Beads were washed three times with wash buffer (10 mM Tris-HCl pH 7.5, 150 mM NaCl, 0.5 mM EDTA). Lastly, 2× Laemmli sample buffer was added to the beads, boiled at 95℃ for 10 min, and cooled on ice before loading on SDS page gels. This was followed by western blotting.

## Western blot

Western blots were performed as described previously (*Ma and Mayr, 2018*). Imaging was captured on the Odyssey CLx imaging system (Li-Cor). The antibodies used are mouse anti-α-tubulin (Sigma-Aldrich, T9026, RRID:AB_477593), mouse anti-mCherry (Abcam, ab125096, RRID:AB_11133266), rabbit anti-HuR (Millipore, 07-1735, RRID:AB_1977173), IRDye 680RD donkey anti-rabbit IgG secondary antibody (Li-COR Biosciences, 926-68073, RRID:AB_10954442), and IRDye 800CW donkey anti-mouse IgG secondary antibody (Li-COR Biosciences, 926-32212, RRID:AB_621847).

## Recombinant protein purification

mGFP-FUS-TIS was cloned into the bacterial expression vector pET28a, which was a gift from the lab of Dirk Remus (MSKCC). At the N-terminus of mGFP, we added a 6xHis-MBP tag, followed by a Tev protease cleavage site. At the C-terminus of TIS11B, we added a Strep-Tag II (SAWSHPQFEK). The 6xHis-MBP tag was PCR-amplified from pDZ2087 construct (Addgene, #92414) with primers MBP F and MBP R. Full-length 6xHis-MBP was cloned into pET28a backbone with *Xba*I and *Eco*RI restriction sites. mGFP-FUS-TIS-Strep-tag II was PCR-amplified from pcDNA-mGFP-FUS-TIS construct with primers mGFP F and TIS RBD-Strep-tag R. The Strep-tag II sequence was incorporated into primer TIS RBD-Strep-tag R. Full-length mGFP-FUS-TIS-Strep-tag II was cloned into pET28a-6xHis-MBP backbone with *Nhe*I and *Eco*RI restriction sites.

To purify high-quality FUS-TIS protein, we used three steps of purification. Step 1: His-Ni purification; step 2: Strep-Tag II purification; and step 3: size exclusion chromatography. pET28a-6xHis-MBP-Tev cleavage site-mGFP-FUS-TIS-Strep-tag II was transformed into BL21 *Escherichia coli* (New England Biolabs). Two fresh colonies were cultivated overnight in 2 × 50 ml SOB medium at 37℃, and then 4 × 25 ml bacteria were transferred to 4 × 1 liter SOB medium to grow at 37℃ until

OD600 reached 0.6. Bacteria were then kept in a 4°C cold room until 18:00. Protein expression was induced by addition of 1 mM IPTG, and bacteria were incubated at 16°C overnight.

Bacteria were centrifuged at 6000 *g* for 10 min, and the pellet was resuspended in 100 ml cold lysis buffer. High-salt lysis buffer (1 M NaCl, 25 mM Tris-Cl, pH 8.0, 20 mM imidazole, 1 mM DTT, 1× PMSF) was used to remove nucleic acid contamination. Bacteria were sonicated on ice for 60 min with on/off interval of 1 and 2 s. The lysate was centrifuged at 13,000 *g* for 30 min.

A 6 ml Ni-NTA (Qiagen) was washed with five column volumes of wash buffer 1 (150 mM NaCl, 25 mM Tris-Cl, pH 8.0, and 1 mM DTT). After centrifugation, the supernatant of the bacteria lysate was transferred into new 50 ml Falcon tubes and incubated with Ni-NTA (Qiagen) at 4°C for 30 min. Then, the sample was transferred into three gravity columns and washed respectively with 40 ml wash buffer 2 (1 M NaCl, 25 mM Tris-Cl, pH 8.0, 20 mM imidazole, and 1 mM DTT), followed with 10 ml wash buffer 3 (600 mM NaCl, 25 mM Tris-Cl, pH 8.0, 20 mM imidazole, and 1 mM DTT). Then, the sample was eluted with 30 ml elution buffer 1 (600 mM NaCl, 25 mM Tris-Cl, pH 8.0, 200 mM imidazole, and 1 mM DTT).

After Ni-NTA purification, the eluted sample was transferred to a 5 ml StrepTrap column (GE Healthcare, cat. no. 28907547), which was pre-equilibrated with elution buffer 2 (600 mM NaCl, 20 mM Tris-HCl, pH 7.4, and 1 mM DTT) using the AKTA Purifier system (GE Healthcare). The target protein was eluted with 20 ml elution buffer 3 (600 mM NaCl, 20 mM Tris-HCl, pH 7.4, 2.5 mM desthiobiotin [Sigma-Aldrich], and 1 mM DTT).

The eluted protein was concentrated using Amicon Ultra-centrifugal filters-50K (Millipore). The concentrated sample was further purified by gel filtration on HiLoad 16/600 Superdex200 column (GE Healthcare) in elution buffer 2 (600 mM NaCl, 20 mM Tris-HCl, pH 7.4, and 1 mM DTT) using the AKTA Purifier system (GE Healthcare).

The fractions representing the monomeric protein were collected and concentrated with Amicon Ultra-centrifugal filters-50K (Millipore). The quality of the final protein product was examined by SDS PAGE. The OD260/280 ratio of the final protein product was 0.52, measured with NanoDrop. The protein concentration was measured by Bradford assay (Bio-Rad). The protein was aliquoted into PCR tubes and flash-frozen in liquid nitrogen and then stored at −80°C.

## In vitro transcription of RNA

All RNAs were in vitro transcribed using the T7 MEGAscript kit (Ambion by Life Technologies). All DNA templates used for in vitro transcription were PCR-amplified and purified with a gel extraction kit (Qiagen). The T7 promoter (TAATACGACTCACTATAGGG) was incorporated into the forward primers used to amplify the DNA templates. After in vitro transcription, products were DNase-treated and run on agarose gels to evaluate the integrity and size of the RNA.

The DNA sequences from the full-length 3′UTRs of *CD47*, *ELAVL1*, *CD274*, and *FUS* were PCR-amplified from pcDNA-eGFP-CD47-3′UTR, eGFP-ELAVL1-3′UTR, eGFP-CD274-3′UTR, and eGFP-FUS-3′UTR constructs. The DNA sequences of the 3′UTRs of *CD44*, *VSIG10*, *IL10*, *TNFSF11*, *GPR39*, *TLR8*, *GPR34*, *TNFAIP6*, *MYC*, *PLA2G4A*, *HEATR5B*, *PPP1R3F*, *DRD1*, *FAM72B*, *MCOLN2*, *TSPAN13*, *LHFPL6*, *FAM174A*, *VPS29*, *ADPGK*, *ASPN*, *CASP8*, *CLCA2*, *EOMES*, *ESCO1*, *GLYATL3*, *HNRNPH3*, *HOGA1*, *LPAR4*, *LRBA*, *LYPLAL1*, *ODF2*, *PRKDC*, *RHOA*, *SHQ1*, *SLC39A6*, *SLC5A9*, *SMIM3*, *SNTN*, *SOSTDC1*, *STBD1*, *TP53TG3*, and *TTC17* were PCR-amplified from HeLa genomic DNA.

To generate the *TNFSF11* 3′UTR mutant carrying two 15-nt oligo insertions, two overlapping fragments were PCR-amplified. Fragment 1: *TNFSF11* 3′UTR with oligo 1 using primers *TNFSF11 3′UTR T7 F* and *TNFSF11* mutant R1. Fragment 2: *TNFSF11* 3′UTR with oligo 1 and oligo 2 using primers *TNFSF11* mutant F2 and *TNFSF11* mutant R2. The inserted 15-nt oligo 2 sequence was incorporated into the primer *TNFSF11* mutant R2. A final ligation PCR was performed to ligate two PCR fragments to generate the full-length *TNFSF11* 3′UTR mutant with primers *TNFSF11 3′UTR T7 F* and *TNFSF11* mutant R2.

RNAs with exogenous dimerization elements (D1a, crRNA, D1b, tracrRNA, D2, HIV dimerization motif) were generated as follows. For D1a-*TLR8*-D2, two rounds of PCR were performed. Round 1: primers D1a-TLR8-D2 1F and D1a-TLR8-D2 1R; round 2: primers D1a-TLR8-D2 2F and D2 R. For D1b-*MYC*-D2, three rounds of PCR were performed. Round 1: primers D1b-MYC-D2 1F and MYC 3′UTR R; round 2: primers D1b-MYC-D2 2F and D1b-MYC-D2 2R; and round 3: primers D1b-MYC-D2 3F and D2 R.

In vitro transcription was performed in a 20 µl volume according to the manufacturer's guidelines. To generate Cy3- or Cy5-labeled RNA, 0.2 µl of 2.5 mM Cy3-UTP or Cy5-UTP (Enzo Life Sciences) was added into the in vitro transcription reaction.

The transcription reaction was incubated 3 hr at 37°C in a PCR machine. All transcribed RNAs were digested with DNase for 30 min at 37°C, then precipitated with LiCl for 4 hr to overnight at −20°C. RNAs were centrifuged at 13,000 for 15 min, and the RNA pellets were washed with 70% ethanol three times. RNAs were dissolved in nuclease-free water and stored at −20°C. The concentration of RNAs was measured by NanoDrop One.

## In vitro phase separation assay

To allow phase separation, purified 6xHis-MBP-mGFP-FUS-TIS-Strep tag II protein stock was incubated with Tev protease for 1 hr at room temperature (RT) to cleave off the 6xHis-MBP tag. mGFP and Strep tag II were not cleaved off. All phase separation assays were performed in 20 µl phase separation buffer (150 mM NaCl, 200 µM $ZnCl_2$, 25 mM Tris-Cl, pH 7.4, 1 mM DTT, 2.5% glycerol, 5% dextran T500 [Pharmacosmos]). Without 5% dextran, there is no phase separation of FUS-TIS protein at the concentration of 10 µM. $ZnCl_2$ was added as the RBD of TIS11B has two zinc finger motifs. Only in the phase separation assay shown in *Figure 2—figure supplement 1D, E* ZnCl2 was omitted.

After 1 hr of Tev protease digestion, the FUS-TIS protein stock was diluted into the desired concentrations with protein stock buffer (600 mM NaCl, 25 mM Tris-Cl, pH 7.4, 1 mM DTT) and centrifuged at 13,000 *g* for 2 min to remove small protein aggregates. The supernatant was transferred into a new Eppendorf tube. The phase separation assay was mixed in PCR tubes. Dextran buffer and RNAs with desired concentrations were first mixed in PCR tubes, then FUS-TIS protein was added into the PCR tube and immediately mixed thoroughly. The final concentrations of FUS-TIS and RNAs are indicated in the figures. The mixture (20 µl) was then transferred into a 384-well glass-bottom microplate (Greiner Bio-One). The chambers of the microplate were pre-treated with 1 mg/ml bovine serum albumin (BSA) (NEB) for 30 min before aspirating the BSA. The microplate was kept in the dark at RT for 2 or 16 hr, followed by imaging of the condensates using confocal microscopy. All phase separation experiments were performed at least three times.

For RNA-only phase separation experiments, Cy5-labeled RNA was diluted with the same phase separation buffer into desired concentrations in a 20 µl volume. The mixture (20 µl) was then transferred into a 384-well glass-bottom microplate. The chambers of the microplate were pre-treated with 1 mg/ml BSA for 30 min before aspirating the BSA. The microplate was kept in the dark at RT for 16 hr, followed by imaging of the Cy5-labeled RNA using confocal microscopy.

## Confocal microscopy

Confocal imaging was performed using ZEISS LSM 880 with Airyscan super-resolution mode. Z stack images were captured with an interval size of 487 nm. A Plan-Apochromat 63x/1.4 Oil objective (Zeiss) was used. For live-cell imaging, HeLa cells were plated on 3.5 cm glass-bottom dishes (Cellvis) and transfected with the indicated constructs. Fourteen hours after transfection, cells were imaged in cell culture medium while incubating in a LiveCell imaging chamber (Zeiss) at 37°C and 5% $CO_2$. Images were prepared with the commercial ZEN software black edition (Zeiss).

## Analysis of the morphology of condensates

The morphology of the condensates was scored by two independent scientists who agreed on the classification. To examine the ability of a specific RNA to induce network formation, several RNA concentrations were tested, for instance, for RNAs with a length of ~1000 nt, we tested concentrations from 50 nM (~16 ng/µl) to 750 nM (~250 ng/µl). If an RNA induced network formation within the concentration range, we considered it as network-forming RNA. If an RNA did not induce network formation even at 750 nM concentration, it was considered as a sphere-forming RNA.

## Fluorescence recovery after photobleaching

FRAP experiments were performed with a ZEISS LSM 880 confocal microscope. A Plan-Apochromat 63x/1.4 Oil objective (Zeiss) was used.

For FRAP of phase separation experiments, 10 µM mGFP-FUS-TIS was mixed with specific RNAs to induce condensate formation. Two hours after mixing, an area of diameter = 1 µm was bleached with a 405 nm and 633 nm laser. GFP or Cy5 (in the case of RNA) fluorescence signal was collected over time.

For FRAP in live cells, HeLa cells were plated on 3.5 cm glass-bottom dishes (Cellvis) and transfected with the indicated constructs. Five or sixteen hours after transfection, cells were imaged in cell culture medium while incubating in a LiveCell imaging chamber (Zeiss) at 37°C and 5% $CO_2$. An area of diameter = 1 µm was bleached with a 405 nm laser. GFP or mCherry fluorescence signal was collected over time. For FRAP of TIS granules at 5 hr after transfection, GFP fluorescence signal was only collected for 10 s after bleaching as the TIS granules are highly mobile.

The prebleached fluorescence intensity was normalized to 1, and the signal after bleaching was normalized to the prebleach level.

## RNA-FISH

Custom Stellaris EGFP FISH probes were described previously (*Berkovits and Mayr, 2015*). RNA-FISH was performed as published with slight modifications (*Ma and Mayr, 2018*). HeLa cells were plated on 4-well Millicell EZ silde and transfected with BFP-FUS-TIS and GFP fusion constructs. Fourteen hours after transfection, cells were washed with PBS, fixed with 4% paraformaldehyde for 15 min at RT, and washed twice for 5 min with PBS. PBS was discarded and 1 ml 70% ethanol was added. The slide was kept at 4°C for 8 hr. The 70% ethanol was aspirated, 1 ml wash buffer was added (2× SSC, 10% formamide in RNase-free water), and incubated at RT for 5 min. Hybridization mix was prepared by mixing 10% dextran sulfate, 10% formamide, 2× SSC, 2 mM ribonucleoside vanadyl complex (NEB), 200 µg/ml yeast tRNA, and FISH probe (1:100). To each well, 200 µl hybridization mix was added and hybridized at 37°C overnight. Slides were washed twice for 30 min each with pre-warmed wash buffer (1 ml, 37°C) in the dark, followed by one quick wash with PBST, and then mounted with mounting solution. Images were captured using confocal ZEISS LSM 880 with Airyscan super-resolution mode (*Huff, 2015*).

## Line profile

In order to examine whether specific mRNAs are enriched in the FUS-TIS granule, line profile analysis was performed. Line profiles were generated with FIJI (ImageJ). A straight line was drawn across the FUS-TIS granule, indicated by the arrows shown in the figures. Fluorescence signals along the straight line of FUS-TIS protein and the examined mRNAs were calculated with the plot profile tool in FIJI. The Pearson's correlation coefficient (r) of two fluorescence signals was calculated with Excel.

## RNA native and denaturing agarose gel electrophoresis

RNA native agarose gel electrophoresis was performed as described previously with a few modifications (*Skripkin et al., 1994*). For sphere-forming and network-forming RNAs, RNAs were diluted into 4 µl buffer A (150 mM NaCl, 25 mM Tris-Cl, pH 7.4) to a final concentration of 5 µM (*CLCA2* 1.7 µg/µl, *TLR8* 1.6 µg/µl, *HNRNPH3* 1.9 µg/µl, *LHFPL6* 1.7 µg/µl, *TNFSF11* 1.8 µg/µl, *TSPAN13* 1.7 µg/µl). RNAs were incubated at 95°C for 2 min in a PCR machine and then incubated on ice for 2 min. RNAs were kept at 37°C for 2 hr. Also, 1 µl native agarose gel loading buffer (6× stock: 60% glycerol, 10 mM Tris-Cl, pH 7.4, 0.03% bromophenol blue, and 0.03% xylene cyanol FF) was added into the RNA. A total of 1 µg RNA was loaded into the 1% agarose gel made with the Tris-acetate-EDTA (TAE) buffer for electrophoresis with TAE buffer.

For RNAs containing dimerization elements (*TLR8* 3′UTR, *MYC* 3′UTR, *D1a-TLR8-D2*, *D1b-MYC-D2*), each RNA was diluted into 4 µl buffer A (150 mM NaCl, 25 mM Tris-Cl, pH 7.4) to a final concentration of 2 µM. RNAs were incubated at 95°C for 2 min in a PCR machine and then incubated on ice for 2 min. Also, 2 µl *TLR8* 3′UTR or 2 µl *MYC* 3′UTR were each diluted with 2 µl buffer A. A 2 µl *TLR8* 3′UTR and 2 µl *MYC* 3′UTR were mixed together. Then, 2 µl *D1a-TLR8-D2* or 2 µl *D1b-MYC-D2* were each diluted with 2 µl buffer A. Also, 2 µl *D1a-TLR8-D2* and 2 µl *D1b-MYC-D2* were mixed together. The final concentration of each RNA was 1 µM (*TLR8* 326 ng/µl, *MYC* 154 ng/µl, *D1a-TLR8-D2* 345 ng/µl, *D1b-MYC-D2* 193 ng/µl) in 4 µl buffer A. RNAs were kept at 37°C for 2 hr. A 1 µl native agarose gel loading buffer was added to the RNA. A total of 1 µg RNA was loaded onto the 2% agarose gel made with the Tris-borate-EDTA (TBE) buffer for electrophoresis with TBE buffer.

For denaturing agarose gel electrophoresis, glyoxal was used. RNAs were mixed with 10 μl glyoxal and incubated at 55°C for 60 min and then incubated on ice for 10 min. A 2 μl agarose gel loading buffer was added into the RNA. A total of 1 μg RNA was loaded into the 1% agarose gel made with the TAE buffer for electrophoresis with TAE buffer.

## Calculation of mRNA concentration in HeLa cells

We estimated the concentration of a specific mRNA in mammalian cells is between 8 pM and 8 nM (9.5 pg/μl to 9.5 ng/μl) based on the following assumptions: (1) the volume of a HeLa cell is 2000 μm³; (2) the average length of an mRNA is 3500 nt, which corresponds to an average molecular weight of 1155 kDa; and (3) there are between 10 and 10,000 copies of mRNAs per cell (*Chen et al., 2016*; *Chen et al., 2015*).

## Calculation of NED values

The ensemble diversity of 3′UTR sequences was calculated using the RNAfold software (version: 2.4.14; command line: RNAfold `–MEA` -d2 -p `–infile=<RNA_sequences.fasta> –outfile=<R-NA_sequences.RNAfold.summary>`) (*Hofacker et al., 1994*; *Lorenz et al., 2011*). Only 3′UTRs with a length <7500 nt can be analyzed by RNAfold. As the values for ensemble diversity depend on the sequence length, we calculated the NED by dividing the value of ensemble diversity by the length of the 3′UTR in nucleotides. All values are listed in *Figure 6—source data 1*. For the 47 experimentally tested 3′UTRs, the NED values range from 0.18 to 0.38. Among the RNAs that induced mesh-like condensates (*N* = 28), 75% of them had NED values higher than 0.28, whereas 75% of the RNAs that induced sphere-like condensates had NED values lower than 0.265. We used these cut-offs to identify 3′UTRs with high or low NED values transcriptome-wide. The range of transcriptome-wide NED values is 0–0.44.

## 3′UTR length, number of AU-rich elements, and HuR binding sites in 3′UTRs

The 3′UTR length is the full-length 3′UTR length obtained from Refseq. For counting of AU-rich elements, we only considered the canonical sequence AUUUA. We counted the number of AU-rich elements in annotated 3′UTRs of mRNAs expressed in HeLa cells. All values are listed in *Figure 6—source data 1*.

PAR-CLIP data of HuR were analyzed from two datasets (*Lebedeva et al., 2011*; *Mukherjee et al., 2011*). Processed peak files were downloaded from POSTAR2 (*Zhu et al., 2019*). Peaks were intersected with a bed file containing human 3′UTRs coordinates (hg38) using bedtools (*Quinlan and Hall, 2010*), and the number of CLIP tags that fall into 3′UTRs was counted. The union of CLIP tags was used to categorize the different groups, meaning that the indicated number of CLIP tags was detected in at least one dataset.

## Statistical methods

For all pair-wise comparisons, a two-sided Mann–Whitney test was performed. For comparisons containing more than two groups, a Kruskal–Wallis test was performed. The Pearson's correlation coefficient is reported.

## Acknowledgements

We thank all members of the Mayr lab for helpful discussions and critical reading of the manuscript. We thank Juncheng Wang for suggestions for protein purification and Neil Robertson for the mCherry-FUS-TIS construct. We also thank Bede Portz and James Shorter for attempting the purification of TIS11B and for valuable suggestions on phase separation experiments. This work was funded by the NIH Director's Pioneer Award (DP1-GM123454), the Pershing Square Sohn Cancer Research Alliance, and the NCI Cancer Center Support Grant (P30 CA008748).

## Additional information

### Funding

| Funder | Grant reference number | Author |
|---|---|---|
| NIH Office of the Director | DP1-GM123454 | Christine Mayr |
| National Cancer Institute | P30 CA008748 | Christine Mayr |
| Pershing Square Sohn Cancer Research Alliance | | Christine Mayr |

The funders had no role in study design, data collection and interpretation, or the decision to submit the work for publication.

### Author contributions

Weirui Ma, Conceptualization, Formal analysis, Validation, Investigation, Visualization, Methodology, Writing - original draft, Writing - review and editing; Gang Zhen, Data curation, Formal analysis, Methodology; Wei Xie, Investigation, Methodology; Christine Mayr, Conceptualization, Supervision, Funding acquisition, Writing - original draft, Writing - review and editing

### Author ORCIDs

Christine Mayr (iD) https://orcid.org/0000-0002-7084-7608

### Decision letter and Author response

Decision letter https://doi.org/10.7554/eLife.64252.sa1
Author response https://doi.org/10.7554/eLife.64252.sa2

## Additional files

### Supplementary files

• Transparent reporting form

### Data availability

All raw data are included as supplementary Excel files in the manuscript.

The following previously published dataset was used:

| Author(s) | Year | Dataset title | Dataset URL | Database and Identifier |
|---|---|---|---|---|
| Zhu Y, Xu G, Yang YT, Xu Z, Chen X, Shi B, Xie D, Lu ZJ, Wang P | 2019 | POSTAR2: deciphering the post-transcriptional regulatory logics | http://lulab.life.tsinghua.edu.cn/postar/index.php | POSTAR2, postar |

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
