## [Decision Letter]

**Acceptance summary:**

The manuscript by Ma et al. investigates the filamentous condensates formed by RNA-protein interactions in cells that have been shown to be ER associated. Since most phase transition condensates are spherical and liquid droplets, the morphology of the mesh-like condensates is striking-clearly visually different. By a variety of experimental approaches, both in vitro and in vivo, the mesh-like condensate is found to be likely due to RNA with "disordered" regions that forms an interacting network of RNA strands bound by proteins with multivalent domains.

**Decision letter after peer review:**

Thank you for submitting your article "A multivalent RNA matrix drives formation of mesh-like condensates" for consideration by *eLife*. Your article has been reviewed by three peer reviewers, one of whom is a member of our Board of Reviewing Editors, and the evaluation has been overseen by James Manley as the Senior Editor. The reviewers have opted to remain anonymous.

The reviewers have discussed the reviews with one another and the Reviewing Editor has drafted this decision to help you prepare a revised submission.

In this study Ma et al. investigate the mechanisms influencing the unique morphology of TIS granules. TIS granules are elongated, filamentous condensates that form extensive contacts with the ER. The authors use a combination of in vivo and in vitro reconstitution experiments using a TIS-FUS fusion protein and various RNAs. They find that long unstructured RNAs are best at generating condensates resembling TIS granules. In contrast, RNAs with secondary structure form rounder condensates. The results are very intriguing and provocative and lend support for a key role for RNA in dictating the structure of membraneless organelles.

The reviewers have numerous suggestions for improving the manuscript, almost all of them requiring changes to the text. Most of these changes recommended have to do with clarity but in some cases the reviewers felt that there should be caveats added, for instance as to the existence of the "mesh". Some conclusions should be modified and referencing expanded as suggested. Some validation experiments using mutations and control constructs with the RBD of TS11B are suggested, but are not required, however you may have this additional data or desire to include it.

The authors should determine whether their title is too broad, since the mesh-like nature of the condensates is inferred.

*Reviewer #1:*

The manuscript by Ma et al. investigates the filamentous condensates formed by RNA-protein interactions in cells that have been shown to be ER associated. Since most phase transition condensates are spherical and liquid droplets, the morphology of the mesh like condensates is striking-clearly visually different. By a variety of experimental approaches, both in vitro and in vivo, the mesh like condensate is found to be likely due to RNA with "disordered" regions that forms an interacting network of RNA strands bound by proteins with multivalent domains.

The definition of mRNAs with disordered regions is a relatively new concept. Although mRNA can adopt secondary structures, it is generally considered to be a floppy molecule without much intrinsic structure. The authors use a score they derive from a folding program, the NED, that can distinguish RNAs with higher structure (similar to the circularization score in Aw, et al., 2016). These programs are usually suggestive rather than predictive and it's not particularly clear exactly what constitutes a disordered region. A test of the RNAs with higher NEDs (less structured) gives an encouraging 79% correspondence with mesh formation. RNAs with low NEDs sometimes could form meshes however (Figure 3, F and G). The line between high and low NEDs seems therefore arbitrary, as if another factor is missing that would make it clearer what is needed for network formation. It's not evident exactly how much the protein component contributes to the mesh formation in vivo, but apparently, it's a major contributor, more than the NED would suggest.

And then there's the question as to what all this means. The authors speculate about it in the Discussion (meshes presumably have more surface interactions, spheres are at the minimal available surface). The functional consequences of mesh disruption have been partially addressed experimentally in a previous publication as aiding in the localization and insertion of membrane proteins and perhaps co-translational assembly of protein complexes. This latter possibility is the most exciting, since it is not clear what role mRNA might play in the assembly of multipolypeptide complexes. A search through the databases might yield some physiological relevance for the mRNAs capable of intermolecular hybridization. This would be a major step toward understanding its contribution to cell physiology.

In all, I feel this work advances our insights into mechanisms leading to the spatial organization and compartmentalization in the cell. It is thoroughly done and at times gets into complex experimentation with a lot of variables (I had to read it four times), so will be a bit of a slog for the average reader. Nonetheless, the extra effort yielded new insights (to me at least) about the possibility of mRNA scaffolds controlling cell homeostasis.

*Reviewer #2:*

TIS granules form in various animal cells under physiological conditions. They display liquid-like properties typical of condensates that form during liquid-liquid phase separation and yet have a mesh-like morphology characteristic of condensates that undergo gelation. In their manuscript, Ma and co-authors aim to address how the mesh-like architecture of TIS granules arises in light of their liquid-likes properties. The authors propose that interacting mRNAs that enrich in TIS granules form an underlying network, which drives the irregular shape of TIS granules while allowing rapid exchange of the TIS granule resident proteins. The authors propose that the RNA structure confers the ability of mRNAs to engage in intermolecular RNA:RNA interactions, which then determines the morphology of TIS granules. The authors arrive at this model by examining the ability of in vitro transcribed 3'UTRs of 47 human mRNAs to induce the mesh-like architecture, all of which are proposed TIS11B targets. The authors find that in silico predicted unstructured 3' UTRs predominantly generated mesh-like granules, whereas, structured 3' UTRs tend to generate spherical granules. Based on these results the authors argued that unstructured RNA are more exposed and thus present more sequences amenable for intermolecular RNA:RNA base-pairing, which then drives the mesh-like granule architecture. In contrast, highly structured mRNAs cannot form such mRNA-mRNA interactions (or at least not many), preventing network formation, thus generating spherical condensates.

In general, this manuscript is well written and provides a coherent message. The images are clear. The authors used in vivo and in vitro systems to address their questions. The methods are well-written and the statistics are properly analyzed. The experiment in Figure 6 A-D is strong and provides compelling evidence for the authors' proposed model. In addition, FRAP data in Figure 7F and Figure 7—figure supplement 1B demonstrated that the RNAs within in vitro reconstituted TIS granules exchange less than the TIS-FUS resident protein that forms the granule and lands further support that the RNA could instruct the shape and stability of RNA granules. These observations agree well with previous in vitro and in vivo measurements which demonstrated that RNAs rather than the granule proteins appear as stable components of a condensate. Overall, I don't think that the authors' model is wrong, however I question whether the authors fully demonstrated that intermolecular RNA:RNA interactions are the major if not the only mechanism that determines the shape and biophysical properties of TIS granules as suggested by the authors. The properties of TIS granules have mostly been examined from a protein point of view and much less from an RNA point of view and mostly in vitro and the evidence of sequence-driven intermolecular RNA:RNA interactions occurring in vivo is very weak. The evidence of interactions occurring in vitro is not sufficient to conclude that these interactions also form in vivo. Specifically, trans-compensatory mutations, which would perturb intermolecular base-pairing and RNA structure are missing. Co-localization data of wild type and mutated interacting TIS11B target mRNAs in cells are also missing. Because of this and possible alternative models that could explain the data, it is difficult to unequivocally conclude that the intermolecular RNA:RNA interactions occur in vivo and that they are responsible for the mesh-like architecture of TIS granules. I have several comments the authors should address before their manuscript is published. While the authors could address these concerns experimentally to bolster their model, I also believe that instead most of these concerns could simply involve amendments to the writing. These re-writings would include a possibility of alternative models, and reflect a more suggestive rather than definitive model of how branched TIS granules could form through intermolecular RNA:RNA interactions. My comments are as follows:

The manuscript warrants publication in *eLife* but should have some conclusions/claims rewritten or addressed. The title of the authors' paper is "A multivalent RNA matrix drives the formation of mesh-like condensates". However, the authors did not unequivocally demonstrate that multivalent intermolecular RNA:RNA interactions indeed occur in vivo and that they are responsible for the mesh-like architecture of TIS granules. In addition, alternative models could explain the data (see major comments posted originally). The authors could address these concerns with revisions of their claims and by discussing alternative models rather than providing substantial amount of additional experimentation. I have listed the concerns the authors should address in my initial review. The authors should also define the NED value better (see Figure 3). These are computational experiments and could be done quickly and should be included in the revised manuscript. Specifically, the authors should address if the NED values in Figure 3 are for 3' UTR or full-length RNA? Are the structures and NEDs of 3' UTRs shown in Figure 3D, E and G consistent with those 3' UTRs from the full-length RNAs? Are the mRNAs, which have unstructured 3'UTR also unstructured in full length? Finally, in Figure 3F, the authors should define the upper and lower limit of NED. Currently, it is difficult to judge whether the NEDs shown present a big or small difference. The authors should predict the NED for a very structured RNA, such as 24 MS2 repeats RNA or a tRNA as well as a linear RNA polymer, such as polyA or polyC RNA.

1) Evidence of pervasive mRNA-mRNA interactions occurring in vivo is missing. Based on the models in Figures 7E and 6D, the authors define intermolecular mRNA-mRNA interactions when one mRNA base pairs with another and propose possible RNA sequences that could base pair. However, this model was never tested. The dimerization experiments using in vitro RNA oligomerization assays revealed that RNAs indeed pair with each other in vitro (Figure 6) but whether this also occurs in vivo is unknown. To test their model, the authors should mutate RNA sequences in TLR8 and MYC to prevent dimerization and test the effect of these mutations on dimerization in vitro and in vivo and to test the effect these sequences have on the granule morphology. The authors could achieve this by using trans-compensatory mutations and mRNA co-localization experiments in vivo (see (Jambor et al., 2011, Ferrandon et al., 1997) for the outline of these experiments). In addition, using FISH, the authors should demonstrate that wild type, interacting mRNAs co-localize in cells while mutated mRNA don't. This should be done with interacting TIS11B target mRNAs.

2) The mesh-like architecture could be driven by a conformational change in the TIS11B protein induced by RNA-binding rather than by intermolecular RNA:RNA interactions. Specifically, the 3'UTRs of TIS11B target mRNAs CD47, ELAV and CD274 induce mesh-like structure of granules while the 3'UTR of FUS, which is not a target mRNAs of TIS11B does not (Figure 2A). However, 3'UTR of FUS is still recruited to the granule, even though it does not appear to be a TIS11B target. Thus, binding of the RNA binding domain (RBD) of TIS11B to its true target could induce a conformational change in TIS11B, which could then induce condensation and branching. A similar conformation-based model was recently proposed for the phase separation of the stress granule protein G3BP1/2 (Yang et al., 2020). This becomes more problematic when the authors test additional 3'UTRs belonging to 47 different human mRNAs. These RNAs all contain multiple AU-rich elements and are considered targets of TIS11B. I had a hard time finding data in the authors' current manuscript or their previous work, which demonstrates that these 47 3'UTRs are bona fide TIS11B targets. It is therefore difficult to determine whether specific binding of TIS11B RBD to true targets induces a conformational change in the protein, which then triggers the mesh-like behavior of granules or whether granule branching is indeed achieved through dimerizing RNAs.

3) The effect of RNA structure on intermolecular RNA:RNA interactions. RNA structure of 3'UTRs predicted in silico drives sphere or mesh-like appearance of TIS granules in vitro. More structured mRNAs cause spherical formation of the granules while less structured RNA tend to trigger mesh-like networks. As the authors propose, this could be due to availability of RNA sequences for intermolecular RNA:RNA interactions. They are exposed in unstructured mRNAs and hidden in structured ones. However, if I follow the point 2 I raised above, then the secondary structure could also hide RNA sequences in target mRNAs that TIS11B protein binds to regardless of how many predicted AU-rich sequences that 3'UTR might have. Conversely, less structured mRNAs could have more exposed sequences for TIS11B binding, which would influence the architecture of TIS granules. This model would also explain the lack of correlation between the number of AU elements and the morphology of granules (Figure 3—figure supplement 5).

4) Reconstituted SUMO-SIM-TIS and FUS-TIS granules do not recapitulate the biophysical properties of TIS11B granules and full length TIS11B granules appear hydrogel-like. The data demonstrate that the tagged full-length TIS11B forms granules that display hydrogel-like rather than liquid-like properties. Specifically, using FRAP, the authors show that a transiently transfected full-length (FL) TIS11B fused with mCherry forms mesh-like condensates reminiscent of the endogenous TIS11B protein. They determine that 16h after of transfection only 20% of TIS11B-GFP exchanged within 120 second. This slow exchange is characteristic of gel-like condensates rather than predominantly liquid-like condensates. This measurement is in contrast to the authors' statement where they indicated that TIS granules are "liquid-like" and that the granule resident proteins are "highly mobile". In addition, to examine granules, the authors chose to work with fluorescently tagged chimeric FUS and SUMO-SIM proteins fussed with the RBD of TIS11B. In contrast to full-length TIS11B, these proteins form mesh-like granules that appear liquid-like; 75% of FUS-TIS exchanges in 25 seconds while close to 100% of SUMO-SIM fully exchanges in just a few seconds (Figure 1E). These data indicate that the chimeras did not recapitulate the basic behavior of TIS granules. In addition, the mCherry-TIS11B protein has a different recovery than mGFP-TIS11B, with the latter one recovering to ~ 43% in 10s (Figure 1—figure supplement 1A). The authors should address these discrepancies in recovery rates and whether TIS-granules are more gel-like instead of liquid-like. Finally, the rationale for switching from tagged FL TIS11B to FUS-TIS and SUMO-SIM-TIS is not clear particularly given that these chimeras do not recapitulate the behavior of FL TIS11B. It is therefore difficult to assess the effect of mRNAs on the biophysical properties of TIS granule resident proteins using a system that behaves fundamentally different than the endogenous system.

*Reviewer #3:*

In this study Ma et al. investigate the mechanisms influencing the unique morphology of TIS granules. TIS granules are elongated, filamentous condensates that form extensive contacts with the ER. The authors use a combination of in vivo and in vitro reconstitution experiments using a TIS-FUS fusion protein and various RNAs. They find that long unstructured RNAs are best at generating condensates resembling TIS granules. In contrast, RNAs with secondary structure form rounder condensates. The results are very intriguing and provocative and lend support for a key role for RNA in dictating the structure of membraneless organelles.

The mechanisms underlying the filamentous structure of TIS-FUS granules remains unclear. The authors consider the possibility that the filaments may be assemblies of smaller condensates that stick rather than fuse with each other, forming elongated "beads on a string". This possibility could be tested by observing the morphology of the condensates at earlier time points. The authors may also want to refer to a recent theoretical study that explains how interactions between multivalent molecules can cause condensates to become kinetically arrested, i.e. unable to grow or fuse as the valency of molecules inside the condensates reach saturation (https://elifesciences.org/articles/56159). This theory may provide a good framework for the authors' observations.

1) An important question addressed by this study is how non-spherical condensates are formed from dynamic protein components. Accordingly, the authors use FRAP experiments to conclude that TIS11 behaves dynamically in vivo. However, only 40% of TIS11 signal recovers after photobleaching (Figure 1—figure supplement 1A). The authors should discuss whether this observation supports conventional liquid-like behavior, or whether a fraction of TIS11 in condensates may be non-dynamic. Stable assembly of TIS11 in vivo raises the possibility that TIS11 protein may contribute to mesh-like condensate assembly.

2) In many cases, the authors base their conclusions on observations made with transfected/tagged TIS11 proteins. Whether these conclusions apply to the native protein is not discussed (is endogenous TIS11 dynamic?). Also the authors should refrain from making sufficiency conclusions based solely on the behavior of transfected proteins. For example, – the TIS11 RNA binding domain is sufficient to make mesh-like condensates in vivo – what is the evidence that this domain is generating a condensate and not just simply localizing to a pre-existing structure?

3) The authors should include the control : GFP-FUS-Strep lacking the TIS11 RBD to ensure that the observed condensates require the TIS11 RBD.

4) The authors should note clearly in the text and figures that the crowding agent dextran is included in all in vitro phase separation assays. They might also comment on the behavior of their construct in the absence of 5% dextran.

---

## [Author Response]

Reviewer #1:The manuscript by Ma et al. investigates the filamentous condensates formed by RNA-protein interactions in cells that have been shown to be ER associated. Since most phase transition condensates are spherical and liquid droplets, the morphology of the mesh like condensates is striking-clearly visually different. By a variety of experimental approaches, both in vitro and in vivo, the mesh like condensate is found to be likely due to RNA with "disordered" regions that forms an interacting network of RNA strands bound by proteins with multivalent domains.The definition of mRNAs with disordered regions is a relatively new concept. Although mRNA can adopt secondary structures, it is generally considered to be a floppy molecule without much intrinsic structure. The authors use a score they derive from a folding program, the NED, that can distinguish RNAs with higher structure (similar to the circularization score in Aw, et al., 2016). These programs are usually suggestive rather than predictive and it's not particularly clear exactly what constitutes a disordered region. A test of the RNAs with higher NEDs (less structured) gives an encouraging 79% correspondence with mesh formation. RNAs with low NEDs sometimes could form meshes however (Figure 3, F and G). The line between high and low NEDs seems therefore arbitrary, as if another factor is missing that would make it clearer what is needed for network formation. It's not evident exactly how much the protein component contributes to the mesh formation in vivo, but apparently, it's a major contributor, more than the NED would suggest.

We thank the reviewer for spending time to review our manuscript and for the insightful comments. For the 47 experimentally tested 3′UTRs, the NED values range from 0.18 – 0.38. Among the RNAs that induced mesh-like condensates (N = 28) 75% of them had NED values higher than 0.28, whereas 75% of the RNAs that induced sphere-like condensates had NED values lower than 0.265. We used these cut-offs to identify 3′UTRs with high or low NED values transcriptome-wide. The range of transcriptome-wide NED values is 0 – 0.44.

We added the information to the text in the Results as well as to the Materials and methods.

We agree that we do not know how much the protein component contributes as we only tested several multivalent domains. It is unclear if most or only a small fraction of proteins can form mesh-like condensates. But it is likely that the ability of proteins to form mesh-like condensates is not limited to a specific multivalent domain, as various multivalent proteins, when bound to RNA, including TIS11B, FXR1, FUS-IDR, SUMO-SIM and PR peptides form mesh-like condensates.

And then there's the question as to what all this means. The authors speculate about it in the Discussion (meshes presumably have more surface interactions, spheres are at the minimal available surface). The functional consequences of mesh disruption have been partially addressed experimentally in a previous publication as aiding in the localization and insertion of membrane proteins and perhaps co-translational assembly of protein complexes. This latter possibility is the most exciting, since it is not clear what role mRNA might play in the assembly of multipolypeptide complexes. A search through the databases might yield some physiological relevance for the mRNAs capable of intermolecular hybridization. This would be a major step toward understanding its contribution to cell physiology.

We currently do not have direct experimental evidence to show the biological relevance of mesh-like condensates. Before our study, mesh-like condensates were largely viewed as aggregates. We hope that our study will motivate more and more people to incorporate mesh-like condensates into their thinking and into their experimental approaches.

In all, I feel this work advances our insights into mechanisms leading to the spatial organization and compartmentalization in the cell. It is thoroughly done and at times gets into complex experimentation with a lot of variables (I had to read it four times), so will be a bit of a slog for the average reader. Nonetheless, the extra effort yielded new insights (to me at least) about the possibility of mRNA scaffolds controlling cell homeostasis.Reviewer #2:[…] My comments are as follows:The manuscript warrants publication in eLife but should have some conclusions/claims rewritten or addressed. The title of the authors' paper is "A multivalent RNA matrix drives the formation of mesh-like condensates". However, the authors did not unequivocally demonstrate that multivalent intermolecular RNA:RNA interactions indeed occur in vivo and that they are responsible for the mesh-like architecture of TIS granules. In addition, alternative models could explain the data (see major comments posted originally). The authors could address these concerns with revisions of their claims and by discussing alternative models rather than providing substantial amount of additional experimentation. I have listed the concerns the authors should address in my initial review. The authors should also define the NED value better (see Figure 3). These are computational experiments and could be done quickly and should be included in the revised manuscript. Specifically, the authors should address if the NED values in Figure 3 are for 3' UTR or full-length RNA? Are the structures and NEDs of 3' UTRs shown in Figure 3D, E and G consistent with those 3' UTRs from the full-length RNAs? Are the mRNAs, which have unstructured 3'UTR also unstructured in full length? Finally, in Figure 3F, the authors should define the upper and lower limit of NED. Currently, it is difficult to judge whether the NEDs shown present a big or small difference. The authors should predict the NED for a very structured RNA, such as 24 MS2 repeats RNA or a tRNA as well as a linear RNA polymer, such as polyA or polyC RNA.

We thank the reviewer for spending the time to review our manuscript and for the insightful comments. We have discussed alternative models in the revised manuscript.

Based on the reviewer’s suggestion, we changed the title. The new title is: in vivo reconstitution finds multivalent RNA-RNA interactions as drivers of mesh-like condensates.

NED values are for 3′UTRs. We used 3′UTRs in our study as they are not protected by ribosomes. We found a weak correlation between the NED values of 3′UTRs and NED values of the full-length mRNAs (Pearson R = 0.17; N = 7,455; P = 3x10^49^). For the 47 experimentally tested 3′UTRs, the NED values range from 0.18 – 0.38. The range of transcriptome-wide NED values is 0 – 0.44 (Figure 6—figure supplement 1). We have added this information to the Materials and methods.

As suggested, we predicted the NED for known structured RNAs. The NED value of tRNA and 6xMS2 are 0.04 and 0.17 respectively, which is consistent with their strong local secondary structures. We added some context to the Results when we describe the NED.

We also predicted the NED values of polyA or polyC RNA, both are 0. This is probably because polyA or polyC cannot form any intramolecular base pairing. We did not include this in the manuscript.

1) Evidence of pervasive mRNA-mRNA interactions occurring in vivo is missing. Based on the models in Figures 7E and 6D, the authors define intermolecular mRNA-mRNA interactions when one mRNA base pairs with another and propose possible RNA sequences that could base pair. However, this model was never tested. The dimerization experiments using in vitro RNA oligomerization assays revealed that RNAs indeed pair with each other in vitro (Figure 6) but whether this also occurs in vivo is unknown. To test their model, the authors should mutate RNA sequences in TLR8 and MYC to prevent dimerization and test the effect of these mutations on dimerization in vitro and in vivo and to test the effect these sequences have on the granule morphology. The authors could achieve this by using trans-compensatory mutations and mRNA co-localization experiments in vivo (see (Jambor et al., 2011, Ferrandon et al., 1997) for the outline of these experiments). In addition, using FISH, the authors should demonstrate that wild type, interacting mRNAs co-localize in cells while mutated mRNA don't. This should be done with interacting TIS11B target mRNAs.

In our in vitro and in vivo reconstitution experiment, the engineered RNAs have three degrees of valency (TLR8-MYC interaction, interaction of D1-D1, and interaction of D2-D2) to enable multivalent RNA-RNA interactions. We used two well-studied dimerization elements: the HIV dimerization motif and the tracrRNA/crRNA dimerization motifs from the CRISPR system. Both dimerization motifs were reported in the literature to dimerize in vivo (Paillart et al., 2004)(Khan et al., 2019).

To show the importance of RNA multivalency in our reconstitution experiment we disrupted the multivalent RNA-RNA interactions. To do so, we removed the dimerization elements D1 and D2. We showed both in vitro and in vivo that reduction of valency (lack of D1 and D2) leads to a dramatic decrease in the ability to induce formation of mesh-like condensates. This result supports our conclusion that multivalent RNA-RNA interactions are required to induce formation of mesh-like condensates.

In our opinion, this experiment is equivalent to mutating the TLR8 or MYC mRNAs as in our manuscript, we only emphasize the requirement of multivalent RNA interactions. We do not focus on specific mRNAs or particular sequence-specific RNA-RNA interactions.

2) The mesh-like architecture could be driven by a conformational change in the TIS11B protein induced by RNA-binding rather than by intermolecular RNA:RNA interactions. Specifically, the 3'UTRs of TIS11B target mRNAs CD47, ELAV and CD274 induce mesh-like structure of granules while the 3'UTR of FUS, which is not a target mRNAs of TIS11B does not (Figure 2A). However, 3'UTR of FUS is still recruited to the granule, even though it does not appear to be a TIS11B target. Thus, binding of the RNA binding domain (RBD) of TIS11B to its true target could induce a conformational change in TIS11B, which could then induce condensation and branching. A similar conformation-based model was recently proposed for the phase separation of the stress granule protein G3BP1/2 (Yang et al., 2020). This becomes more problematic when the authors test additional 3'UTRs belonging to 47 different human mRNAs. These RNAs all contain multiple AU-rich elements and are considered targets of TIS11B. I had a hard time finding data in the authors' current manuscript or their previous work, which demonstrates that these 47 3'UTRs are bona fide TIS11B targets. It is therefore difficult to determine whether specific binding of TIS11B RBD to true targets induces a conformational change in the protein, which then triggers the mesh-like behavior of granules or whether granule branching is indeed achieved through dimerizing RNAs.

We thank the reviewer for the insightful comment. It is possible that the formation of mesh-like morphology is driven by a conformational change in the RNA-binding protein. In our opinion, this model cannot fully explain our experimental results. This model would only work if all by chance all network-forming RNAs are true targets and all sphere-forming RNAs are not true targets despite having the similar numbers of AU-rich elements. AU-rich elements are the binding sites for TIS11B, but their presence does not prove that TIS11B actually binds. In the meantime, we obtained iCLIP data for TIS11B and found no difference in the number of iCLIP tags in sphere-forming vs network-forming RNAs.

Besides, in our in vivo reconstitution experiment, we use the MS2 coat protein and SUMO-SIM, and mRNAs are recruited to the condensates by MS2 repeats. All RNAs have the same MS2 binding sites and are true targets of MCP-SUMO-SIM. But still, when the RNA dimerization motifs D1 and D2 were eliminated, the ability to induce mesh-like condensate formation is significantly reduced compared to RNAs that contain the D1 and D2 elements. Therefore, we do not think that our data are better explained by a model that infers a conformational change in the protein.

For TIS granules, our transcriptome-wide analysis showed that TIS11B target mRNAs have a high propensity to form multivalent RNA-RNA interactions as they are enriched for large disordered regions and have high NED values. It is likely that multivalent RNA-RNA interactions are responsible for the mesh-like morphology of TIS granules. Nevertheless, we agree with the reviewer that we do not have direct evidence to show that multivalent RNA-RNA interactions happen in TIS granules. We cannot exclude the possibility that the mesh-like architecture of TIS granules could be driven by a conformational change in the TIS11B protein induced by RNA-binding. It is possible that TIS granules either use RNA interactions or a conformational change, or that both mechanisms contribute to its mesh-like morphology. We have added this alternative model to our manuscript.

3) The effect of RNA structure on intermolecular RNA:RNA interactions. RNA structure of 3'UTRs predicted in silico drives sphere or mesh-like appearance of TIS granules in vitro. More structured mRNAs cause spherical formation of the granules while less structured RNA tend to trigger mesh-like networks. As the authors propose, this could be due to availability of RNA sequences for intermolecular RNA:RNA interactions. They are exposed in unstructured mRNAs and hidden in structured ones. However, if I follow the point 2 I raised above, then the secondary structure could also hide RNA sequences in target mRNAs that TIS11B protein binds to regardless of how many predicted AU-rich sequences that 3'UTR might have. Conversely, less structured mRNAs could have more exposed sequences for TIS11B binding, which would influence the architecture of TIS granules. This model would also explain the lack of correlation between the number of AU elements and the morphology of granules (Figure 3—figure supplement 5).

Both groups of mRNAs have similar numbers of TIS11B binding sites identified by iCLIP, suggesting that similar numbers of binding sites are accessible. See our response to comment 2.

4) Reconstituted SUMO-SIM-TIS and FUS-TIS granules do not recapitulate the biophysical properties of TIS11B granules and full length TIS11B granules appear hydrogel-like. The data demonstrate that the tagged full-length TIS11B forms granules that display hydrogel-like rather than liquid-like properties. Specifically, using FRAP, the authors show that a transiently transfected full-length (FL) TIS11B fused with mCherry forms mesh-like condensates reminiscent of the endogenous TIS11B protein. They determine that 16h after of transfection only 20% of TIS11B-GFP exchanged within 120 second. This slow exchange is characteristic of gel-like condensates rather than predominantly liquid-like condensates. This measurement is in contrast to the authors' statement where they indicated that TIS granules are "liquid-like" and that the granule resident proteins are "highly mobile". In addition, to examine granules, the authors chose to work with fluorescently tagged chimeric FUS and SUMO-SIM proteins fussed with the RBD of TIS11B. In contrast to full-length TIS11B, these proteins form mesh-like granules that appear liquid-like; 75% of FUS-TIS exchanges in 25 seconds while close to 100% of SUMO-SIM fully exchanges in just a few seconds (Figure 1E). These data indicate that the chimeras did not recapitulate the basic behavior of TIS granules. In addition, the mCherry-TIS11B protein has a different recovery than mGFP-TIS11B, with the latter one recovering to ~ 43% in 10s (Figure 1—figure supplement 1A). The authors should address these discrepancies in recovery rates and whether TIS-granules are more gel-like instead of liquid-like. Finally, the rationale for switching from tagged FL TIS11B to FUS-TIS and SUMO-SIM-TIS is not clear particularly given that these chimeras do not recapitulate the behavior of FL TIS11B. It is therefore difficult to assess the effect of mRNAs on the biophysical properties of TIS granule resident proteins using a system that behaves fundamentally different than the endogenous system.

In our previous paper, we performed FRAP experiments after 16 hours of transfection of mCherry-TIS11B and observed that only about 20% of the fluorescent signal recovered in 120 seconds. This observation led to the conclusion that the mesh-like morphology of TIS granules is simply caused by their gel-like nature. Later, we performed FRAP experiments after 5 hours of transfection of GFP-TIS11B and observed that 43% of the fluorescent signal recovered in 10 seconds. At this time point, it is impossible to record longer as TIS granules move very fast in cells. Because of 43% recovery in 10 seconds, we did no longer consider TIS granules as gel-like and we described them in this manuscript as “somewhat dynamic”. At the 5-hour time point the morphology of TIS granules is irregular. Therefore we hypothesized that the mesh-like morphology of TIS granules cannot simply be explained by gel-like biophysical properties.

We then used chimeric proteins to better understand how the irregularly shaped morphology is generated. When we saw that the mesh-like condensates formed by SUMO-SIM-TIS or FUS-TIS showed fast FRAP recovery (their fluorescence recovered to more than 75% and 90% within 30 seconds of bleaching), we became very excited. Based on the FRAP data, SUMO-SIM-TIS and FUS-TIS condensates would be considered to have liquid-like properties. However, based on the current knowledge this is a paradox as due to surface tension, liquid-like condensates should have a sphere-like morphology and should not be mesh-like.

At that moment, we changed the focus of our study. Instead of studying how the mesh-like morphology of TIS granules is generated, we rather set out to investigate the paradoxical observation how liquid-like condensates can have a mesh-like morphology.

In our opinion this is a more general question than the questions of what causes the mesh-like morphology of TIS granules, for which our experimental tools were limited at the time as we only had constructs in hands that poorly mimic the behavior of endogenous TIS granules. We would like to point out that this manuscript is not primarily about TIS granules, but rather about finding ways to explain how mesh-like condensates with dynamic protein components are formed. We did find one such mechanism: Multivalent RNA-RNA interactions can act as skeleton of mesh-like condensates. We are not saying that this is the only mechanism as there are likely other mechanisms that induce mesh-like condensates with dynamic protein components.

We added a more extensive explanation on the rationale for the use of chimeric proteins to the text.

Reviewer #3:In this study Ma et al. investigate the mechanisms influencing the unique morphology of TIS granules. TIS granules are elongated, filamentous condensates that form extensive contacts with the ER. The authors use a combination of in vivo and in vitro reconstitution experiments using a TIS-FUS fusion protein and various RNAs. They find that long unstructured RNAs are best at generating condensates resembling TIS granules. In contrast, RNAs with secondary structure form rounder condensates. The results are very intriguing and provocative and lend support for a key role for RNA in dictating the structure of membraneless organelles.The mechanisms underlying the filamentous structure of TIS-FUS granules remains unclear. The authors consider the possibility that the filaments may be assemblies of smaller condensates that stick rather than fuse with each other, forming elongated "beads on a string". This possibility could be tested by observing the morphology of the condensates at earlier time points. The authors may also want to refer to a recent theoretical study that explains how interactions between multivalent molecules can cause condensates to become kinetically arrested, i.e. unable to grow or fuse as the valency of molecules inside the condensates reach saturation (https://elifesciences.org/articles/56159). This theory may provide a good framework for the authors' observations.

We thank the reviewer for spending the time to review our manuscript and for the insightful comments. We did confocal 3D time-lapse imaging of FUS-TIS condensates in vitro after 30 min of mixing with CD47 3′UTR RNA (network formation) or TLR8 3′UTR RNA (sphere formation). The results showed that smaller FUS-TIS condensates mixed with CD47 3′UTR RNA stick rather than fuse with each other. The data were shown in Figures 7A and 7B. In cells, the morphology of FUS-TIS condensates cannot be observed at an earlier time point as they move very fast when they are small. We thank the reviewer for suggesting to cite this paper. We added it to the new version of the manuscript.

1) An important question addressed by this study is how non-spherical condensates are formed from dynamic protein components. Accordingly, the authors use FRAP experiments to conclude that TIS11 behaves dynamically in vivo. However, only 40% of TIS11 signal recovers after photobleaching (Figure 1—figure supplement 1A). The authors should discuss whether this observation supports conventional liquid-like behavior, or whether a fraction of TIS11 in condensates may be non-dynamic. Stable assembly of TIS11 in vivo raises the possibility that TIS11 protein may contribute to mesh-like condensate assembly.

In our previous paper, we performed FRAP experiments after 16 hours of transfection of mCherry-TIS11B and observed that only about 20% of the fluorescent signal recovered in 120 seconds. This observation led to the conclusion that the mesh-like morphology of TIS granules is simply caused by their gel-like nature. Later, we performed FRAP experiments after 5 hours of transfection of GFP-TIS11B and observed that 43% of the fluorescent signal recovered in 10 seconds. At this time point, it is impossible to record longer as TIS granules move very fast in cells. Because of 43% recovery in 10 seconds, we did no longer consider TIS granules as gel-like and we described them in this manuscript as “somewhat dynamic”. At the 5-hour time point the morphology of TIS granules is irregular. Therefore we hypothesized that the mesh-like morphology of TIS granules cannot simply be explained by gel-like biophysical properties.

We then used chimeric proteins to better understand how the irregularly shaped morphology is generated. When we saw that the mesh-like condensates formed by SUMO-SIM-TIS or FUS-TIS showed fast FRAP recovery (their fluorescence recovered to more than 75% and 90% within 30 seconds of bleaching), we became very excited. Based on the FRAP data, SUMO-SIM-TIS and FUS-TIS condensates would be considered to have liquid-like properties. However, based on the current knowledge this is a paradox as due to surface tension, liquid-like condensates should have a sphere-like morphology and should not be mesh-like.

At that moment, we changed the focus of our study. Instead of studying how the mesh-like morphology of TIS granules is generated, we rather set out to investigate the paradoxical observation how liquid-like condensates can have a mesh-like morphology.

In our opinion this is a more general question than the questions of what causes the mesh-like morphology of TIS granules, for which our experimental tools were limited at the time as we only had constructs in hands that poorly mimic the behavior of endogenous TIS granules. We would like to point out that this manuscript is not primarily about TIS granules, but rather about finding ways to explain how mesh-like condensates with dynamic protein components are formed. We did find one such mechanism: Multivalent RNA-RNA interactions can act as skeleton of mesh-like condensates. We are not saying that this is the only mechanism as there are likely other mechanisms that induce mesh-like condensates with dynamic protein components.

We added a more extensive explanation on the rationale for the use of chimeric proteins to the text.

2) In many cases, the authors base their conclusions on observations made with transfected/tagged TIS11 proteins. Whether these conclusions apply to the native protein is not discussed (is endogenous TIS11 dynamic?). Also the authors should refrain from making sufficiency conclusions based solely on the behavior of transfected proteins. For example, – the TIS11 RNA binding domain is sufficient to make mesh-like condensates in vivo – what is the evidence that this domain is generating a condensate and not just simply localizing to a pre-existing structure?

As pointed out in the manuscript and in the previous comment, the manuscript does not primarily focus on TIS granules as we currently do not have the tools to assess the material properties of endogenous TIS granules. However, our findings are likely valid for endogenous TIS granules because our transcriptome-wide analyses showed that the TIS11B target mRNAs have higher NED values than other mRNAs and thus, have a higher propensity to form multivalent RNA-RNA interactions. In our opinion, it is very likely that also multivalent RNA-RNA interactions are responsible for the mesh-like morphology of TIS granules.

When the TIS RNA-binding domain was fused to SUMO-SIM or FUS IDR both chimeric proteins formed mesh-like condensates. However, the size and localization of SUMO-SIM-TIS condensates is very different from both FUS-TIS condensates and endogenous TIS granules. Therefore, it is unlikely that they simply localize to endogenous TIS granules. But we cannot exclude the possibility that FUS-TIS and SUMO-SIM-TIS localize to two different pre-existing structures. We have deleted the statement of sufficiency.

3) The authors should include the control : GFP-FUS-Strep lacking the TIS11 RBD to ensure that the observed condensates require the TIS11 RBD.

It was reported by Anthony Hyman’s lab that FUS IDR requires a very high concentration to undergo phase separation. With 10% dextran, a concentration of more than 20 µM is required to induce significant phase separation of the FUS IDR (Jie et al., Cell 2018. Figure S2D). In our experiment, we use 10 µM protein and 5% dextran. With this condition, we expect that the FUS IDR will not undergo phase separation. Therefore, we did not include the GFP-FUS IDR-Strep tag II as a control.

In our in vivo reconstitution experiment (Figure 6), we use the MS2 coat protein (MCP) as an RNA binding domain to recruit specific RNAs. Without MCP or MS2 repeats, 100% SUMO-SIM condensates are sphere-like, which provides another line of evidence to support the point that specific RNAs need to be recruited into the condensates to induce network formation.

4) The authors should note clearly in the text and figures that the crowding agent dextran is included in all in vitro phase separation assays. They might also comment on the behavior of their construct in the absence of 5% dextran.

Without 5% dextran, there is no phase separation of FUS-TIS protein at the concentration of 10 µM. We added this information to the Materials and methods. We also added to the text and to the figure legend of Figure 2 that all phase separation experiments were performed in the presence of 5% Dextran.